# Communication-Efficient Gradient Descent-Accent Methods for Distributed Variational Inequalities: Unified Analysis and Local Updates

**Siqi Zhang**[*]
AMS & MINDS
Johns Hopkins University
szhan207@jhu.edu

**Sayantan Choudhury**[*]
AMS & MINDS
Johns Hopkins University
schoudh8@jhu.edu

**Sebastian U Stich**
CISPA Helmholtz Center for Information Security
stich@cispa.de

**Nicolas Loizou**
AMS & MINDS
Johns Hopkins University
nloizou@jhu.edu

## Abstract

Distributed and federated learning algorithms and techniques associated primarily with minimization problems. However, with the increase of minimax optimization and variational inequality problems in machine learning, the necessity of designing efficient distributed/federated learning approaches for these problems is becoming more apparent. In this paper, we provide a unified convergence analysis of communication-efficient local training methods for distributed variational inequality problems (VIPs). Our approach is based on a general key assumption on the stochastic estimates that allows us to propose and analyze several novel local training algorithms under a single framework for solving a class of structured non-monotone VIPs. We present the first local gradient descent-accent algorithms with provable *improved communication complexity* for solving distributed variational inequalities on heterogeneous data. The general algorithmic framework recovers state-of-the-art algorithms and their sharp convergence guarantees when the setting is specialized to minimization or minimax optimization problems. Finally, we demonstrate the strong performance of the proposed algorithms compared to state-of-the-art methods when solving federated minimax optimization problems.

## 1 Introduction

Federated learning (FL) (Konečný et al., 2016; McMahan et al., 2017; Kairouz et al., 2021) has become a fundamental distributed machine learning framework in which multiple clients collaborate to train a model while keeping their data decentralized. Communication overhead is one of the main bottlenecks in FL (Karimireddy et al., 2020), which motivates the use by the practitioners of advanced algorithmic strategies to alleviate the communication burden. One of the most popular and well-studied strategies to reduce the communication cost is increasing the number of local steps between the communication rounds (McMahan et al., 2017; Stich, 2018; Assran et al., 2019; Khaled et al., 2020a; Koloskova et al., 2020).

Towards a different objective, several recently proposed machine learning approaches have moved from the classical optimization formulation to a multi-player game perspective, where a good model is framed as the equilibrium of a game, as opposed to the minimizer of an objective function. Such machine learning models are framed as games (or in their simplified form as minimax optimization problems) and involve interactions between players, e.g., Generative Adversarial Networks (GANs)

---

[*]Equal contribution.

(Goodfellow et al., 2014), adversarial training (Madry et al., 2018), robust optimization (Namkoong & Duchi, 2016), and multi-agent reinforcement learning (Zhang et al., 2021a).

Currently, federated learning algorithms and techniques are associated primarily with minimization problems. However, with the increase of game-theoretical formulations in machine learning, the necessity of designing efficient federated learning approaches for these problems is apparent. In this work, we are interested in the design of communication-efficient federated learning algorithms suitable for multi-player game formulations. In particular, we consider a more abstract formulation and focus on solving the following distributed/federated variational inequality problem (VIP):

$$\text{Find } z^* \in \mathbb{R}^{d'}, \text{ such that } \langle F(z^*), z - z^* \rangle \geq 0, \quad \forall z \in \mathbb{R}^{d'}, \tag{1}$$

where $F : \mathbb{R}^{d'} \to \mathbb{R}^{d'}$ is a structured non-monotone operator. We assume the operator $F$ is distributed across $n$ different nodes/clients and written as $F(z) = \frac{1}{n}\sum_{i=1}^{n} f_i(z)$. In this setting, the operator $f_i : \mathbb{R}^{d'} \to \mathbb{R}^{d'}$ is owned by and stored on client $i$. Since we are in the unconstrained scenario, problem (1) can be equivalently written as finding $z^* \in \mathbb{R}^{d'}$, such that $F(z^*) = 0$ (Loizou et al., 2021; Gorbunov et al., 2022a).

Problem (1) can be seen as a more abstract formulation of several popular problems. In particular, the selection of the operator $F$ captures classical minimization problems $\min_{z \in \mathbb{R}^d} f(z)$, minimax optimization problems with $z = (x_1, x_2)$, $\min_{x_1 \in \mathbb{R}^{d_1}} \max_{x_2 \in \mathbb{R}^{d_2}} f(x_1, x_2)$, and multi-player games as special cases. For example, in the minimization case, $F(z) = \nabla f(z)$, and it is easy to see that $z^*$ is the solution of (1) if and only if $\nabla f(z^*) = 0$, while in the minimax case $F(z) = F(x_1, x_2) = (\nabla_{x_1} f(x_1, x_2), -\nabla_{x_2} f(x_1, x_2))$ and $z^*$ is the solution if and only if $z^*$ is a stationary point of $f$. More generally, an important special case of the formulation (1) is the problem of finding the equilibrium point in an $N$-players game. In this setting, each player $j$ is simultaneously trying to find the action $x_j^*$ which minimizes with respect to $x_j \in \mathbb{R}^{d_j}$ their own cost function $f_j(x_j, x_{-j})$, while the other players are playing $x_{-j}$, which represents $x = (x_1, \ldots, x_k)$ with the component $j$ removed. In this scenario, operator $F(x)$ is the concatenation over all possible $j$'s of $\nabla_{x_j} f_j(x_j, x_{-j})$.

Finally, in the distributed/federated learning setting, formulation (1) captures both classical FL minimization (McMahan et al., 2017) and FL minimax optimization problems (Sharma et al., 2022; Deng & Mahdavi, 2021) as special cases.

## 1.1 MAIN CONTRIBUTIONS

Our key contributions are summarized as follows:

- **VIPs and Federated Learning:** We present the first connection between regularized VIPs and federated learning setting by explaining how one can construct a consensus reformulation of problem(1) by appropriately selecting the regularizer term $R(x)$ (see discussion in Section 2.1).

- **Unified Framework:** We provide a unified theoretical framework for the design of efficient local training methods for solving the distributed VIP (1). For our analysis, we use a general key assumption on the stochastic estimates that allows us to study, under a single framework, several stochastic local variants of the proximal gradient descent-ascent (GDA) method [1].

- **Communication Acceleration:** In practice, local algorithms are superior in communication complexity compared to their non-local counterparts. However, in the literature, the theoretical communication complexity of Local GDA does not improve upon the vanilla distributed GDA (i.e., communication in every iteration). Here we close the gap and provide the first communication-accelerated local GDA methods. For the deterministic strongly monotone and smooth setting, our method requires $\mathcal{O}\left(\kappa \ln \frac{1}{\epsilon}\right)$ communication rounds over the $\mathcal{O}\left(\kappa^2 \ln \frac{1}{\epsilon}\right)$ of vanilla GDA (and previous analysis of Local GDA), where $\kappa$ is the condition number. See Table 1 for further details .

- **Heterogeneous Data:** Designing algorithms for federated minimax optimization and distributed VIPs, is a relatively recent research topic, and existing works heavily rely on the bounded heterogeneity assumption, which may be unrealistic in FL setting (Hou et al., 2021; Beznosikov et al., 2020; Sharma et al., 2022). Thus, they can only solve problems with similar data between clients/nodes. In practical scenarios, the private data stored by a user on a mobile device and the

---

[1]Throughout this work we use this suggestive name (GDA and Stochastic GDA/ SGDA) motivated by the minimax formulation, but we highlight that our results hold for the more general VIP (1)

Table 1: Summary and comparison of algorithms for solving strongly-convex-strongly-concave federated minimax optimization problems (a special case of the distributed VIPs (1)).

| Algorithm[1] | Acceleration? | Variance Red.? | Communication Complexity |
|---|---|---|---|
| **GDA/SGDA** (Fallah et al., 2020) | — | ✗ | $\mathcal{O}\left(\max\left\{\kappa^2, \frac{\sigma_*^2}{\mu^2 \epsilon}\right\} \ln \frac{1}{\epsilon}\right)$ |
| **Local GDA/SGDA** (Deng & Mahdavi, 2021) | ✗ | ✗ | $\mathcal{O}\left(\sqrt{\frac{\kappa^2(\sigma_*^2 + \Delta^2)}{\mu \epsilon}}\right)$ |
| **ProxSkip-GDA/SGDA-FL** (This work) | ✓ | ✗ | $\mathcal{O}\left(\sqrt{\max\left\{\kappa^2, \frac{\sigma_*^2}{\mu^2 \epsilon}\right\}} \ln \frac{1}{\epsilon}\right)$ |
| **ProxSkip-L-SVRGDA-FL** (This work) | ✓ | ✓ | $\mathcal{O}\left(\kappa \ln \frac{1}{\epsilon}\right)$ |

[1] "Acceleration" = whether the algorithm enjoys acceleration in communication compared to its non-local counterpart, "Variance Red.?" = whether the algorithm (in stochastic setting) applies variance reduction, "Communication Complexity" = the communication complexity of the algorithm. Here $\kappa = L/\mu$ denotes the condition number where $L$ is the Lipschitz parameter and $\mu$ is the modulus of strong convexity, $\sigma_*^2$ corresponds to the variance level at the optimum ($\sigma_*^2 = 0$ in deterministic setting), and $\Delta$ captures the bounded variance. For a more detailed comparison of complexities, please also refer to Table 2.

    data of different users can be arbitrarily heterogeneous. Our analysis does not assume bounded heterogeneity, and the proposed algorithms (ProxSkip-VIP-FL and ProxSkip-L-SVRGDA-FL) guarantee convergence with an improved communication complexity.

- **Sharp rates for known special cases:** For the known methods/settings fitting our framework, our general theorems recover the best rates known for these methods. For example, the convergence results of the Proximal (non-local) SGDA algorithm for regularized VIPs (Beznosikov et al., 2022b) and the ProxSkip algorithm for composite minimization problems (Mishchenko et al., 2022) can be obtained as special cases of our analysis, showing the tightness of our convergence guarantees.

- **Numerical Evaluation:** In numerical experiments, we illustrate the most important properties of the proposed methods by comparing them with existing algorithms in federated minimax learning tasks. The numerical results corroborate our theoretical findings.

## 2 TECHNICAL PRELIMINARIES

### 2.1 REGULARIZED VIP AND CONSENSUS REFORMULATION

Following classical techniques from Parikh & Boyd (2014), the distributed VIP (1) can be recast into a consensus form:

$$\text{Find } x^* \in \mathbb{R}^d, \text{such that } \langle F(x^*), x - x^* \rangle + R(x) - R(x^*) \geq 0, \quad \forall x \in \mathbb{R}^d, \tag{2}$$

where $d = nd'$ and

$$F(x) \triangleq \sum_{i=1}^n F_i(x_i), \quad R(x) \triangleq \begin{cases} 0 & \text{if } x_1 = x_2 = \cdots = x_n \\ +\infty & \text{otherwise.} \end{cases} \tag{3}$$

Here $x = (x_1, x_2, \cdots, x_n) \in \mathbb{R}^d$, $x_i \in \mathbb{R}^{d'}$, $F : \mathbb{R}^d \to \mathbb{R}^d$, $F_i : \mathbb{R}^{d'} \to \mathbb{R}^d$ and $F_i(x_i) = (0, \cdots, f_i(x_i), \cdots, 0)$ where $[F(x)]_{(id'+1):(id'+d')} = f_i(x_i)$. Note that the reformulation requires a dimension expansion on the variable enlarged from $d'$ to $d$. It is well-known that the two problems (1) and (2) are equivalent in terms of the solution, which we detail in Appendix F.3.

Having explained how the problem (1) can be converted into a regularized VIP (2), let us now present the Stochastic Proximal Method (Parikh & Boyd, 2014; Beznosikov et al., 2022b), one of the most popular algorithms for solving general regularized VIPs[2]. The update rule of the Stochastic Proximal Method defines as follows:

$$x_{t+1} = \mathbf{prox}_{\gamma R}(x_t - \gamma g_t) \quad \text{where} \quad \mathbf{prox}_{\gamma R}(x) \triangleq \underset{v \in \mathbb{R}^d}{\arg\min}\left\{R(v) + \frac{1}{2\gamma}\|v - x\|^2\right\}. \tag{4}$$

---

[2] We call general regularized VIPs, the problem (2) where $F : \mathbb{R}^d \to \mathbb{R}^d$ is an operator and $R : \mathbb{R}^d \to \mathbb{R}$ is a regularization term (a proper lower semicontinuous convex function).

Here $g_t$ is an unbiased estimator of $F(x_t)$ and $\gamma > 0$ is the step-size of the method. As explained in (Mishchenko et al., 2022), it is typically assumed that the proximal computation (4) can be evaluated in closed form (has exact value), and its computation it is relatively cheap. That is, the bottleneck in the update rule of the Stochastic Proximal Method is the computation of $g_t$. This is normally the case when the regularizer term $R(x)$ in general regularized VIP has a simple expression, like $\ell_1$-norm ($R(x) = \|x\|_1$) and $\ell_2$-norm ($R(x) = \|x\|_2^2$). For more details on closed-form expression (4) under simple choices of regularizers $R(x)$, we refer readers to check Parikh & Boyd (2014).

However, in the consensus reformulation of the distributed VIP, the regularizer $R(x)$ has a specific expression (3) that makes the proximal computation (4) expensive compared to the evaluation of $g_t$. In particular, note that by following the definition of $R(x)$ in (3), we get that $\bar{x} = \frac{1}{n} \sum_{i=1}^{n} x_i$ and $\mathbf{prox}_{\gamma R}(x) = (\bar{x}, \bar{x}, \cdots, \bar{x})$. So evaluating $\mathbf{prox}_{\gamma R}(x)$ is equivalent to taking the average of the variables $x_i$ (Parikh & Boyd, 2014), meaning that it involves a high communication cost in distributed/federated learning settings. This was exactly the motivation behind the proposal of the ProxSkip algorithm in Mishchenko et al. (2022), which reduced the communication cost by allowing the expensive proximal operator to be skipped in most iterations.

In this work, inspired by the ProxSkip approach of Mishchenko et al. (2022), we provide a unified framework for analyzing efficient algorithms for solving the distributed VIP (1) and its consensus reformulation problem (2). In particular in Section 3, we provide algorithms for solving general regularized VIPs (not necessarily a distributed setting). Later in Section 4 we explain how the proposed algorithms can be interpreted as distributed/federated learning methods.

## 2.2 MAIN ASSUMPTIONS

Having presented the consensus reformulation of distributed VIPs, let us now provide the main conditions of problem (2) assumed throughout the paper.

**Assumption 2.1.** We assume that Problem (2) has a unique solution $x^*$ and

1. The operator $F$ is $\mu$-quasi-strongly monotone and $\ell$-star-cocoercive with $\mu, \ell > 0$, i.e., $\forall x \in \mathbb{R}^d$,

$$\langle F(x) - F(x^*), x - x^* \rangle \geq \mu \|x - x^*\|^2, \quad \langle F(x) - F(x^*), x - x^* \rangle \geq \frac{1}{\ell} \|F(x) - F(x^*)\|^2.$$

2. The function $R(\cdot)$ is a proper lower semicontinuous convex function.

Assumption 2.1 is weaker than the classical strong-monotonicity and Lipschitz continuity assumptions commonly used in analyzing methods for solving problem (2) and captures non-monotone and non-Lipschitz problems as special cases (Loizou et al., 2021). In addition, given that the operator $F$ is $L$-Lipschitz continuous and $\mu$-strongly monotone, it can be shown that the operator $F$ is $(\kappa L)$-star-cocoercive where $\kappa \triangleq L/\mu$ is the condition number of the operator (Loizou et al., 2021).

Motivated by recent applications in machine learning, in our work, we mainly focus on the case where we have access to unbiased estimators of the operator $F$. Regarding the inherent stochasticity, we further use the following key assumption, previously used in Beznosikov et al. (2022b) for the analysis of Proximal SGDA, to characterize the behavior of the operator estimation.

**Assumption 2.2** (Estimator). For all $t \geq 0$, we assume that the estimator $g_t$ is unbiased ($\mathbb{E}[g_t] = F(x_t)$). Next, we assume that there exist non-negative constants $A, B, C, D_1, D_2 \geq 0$, $\rho \in (0, 1]$ and a sequence of (possibly random) non-negative variables $\{\sigma_t\}_{t \geq 0}$ such that for all $t \geq 0$:

$$\mathbb{E}\|g_t - F(x^*)\|^2 \leq 2A\langle F(x) - F(x^*), x - x^* \rangle + B\sigma_t^2 + D_1,$$
$$\mathbb{E}[\sigma_{t+1}^2] \leq 2C\langle F(x) - F(x^*), x - x^* \rangle + (1 - \rho)\sigma_t^2 + D_2.$$

Variants of Assumption 2.2 have first proposed in classical minimization setting for providing unified analysis for several stochastic optimization methods and, at the same time, avoid the more restrictive bounded gradients and bounded variance assumptions (Gorbunov et al., 2020; Khaled et al., 2020b). Recent versions of Assumption 2.2 have been used in the analysis of Stochastic Extragradient (Gorbunov et al., 2022a) and stochastic gradient-descent assent (Beznosikov et al., 2022b) for solving minimax optimization and VIP problems. To our knowledge, analysis of existing local training methods for distributed VIPs depends on bounded variance conditions. Thus, via Assumption 2.2, our convergence guarantees hold under more relaxed assumptions as well. Note that

Assumption 2.2 covers many well-known conditions: for example, when $\sigma_t \equiv 0$ and $F(x^*) = 0$, it will recover the recently introduced expected co-coercivity condition (Loizou et al., 2021) where $A$ corresponds to the modulus of expected co-coercivity, and $D_1$ corresponds to the scale of estimator variance at $x^*$. In Section 3.2, we explain how Assumption 2.2 is satisfied for many well-known estimators $g_t$, including the vanilla mini-batch estimator and variance reduced-type estimator. That is, for the different estimators (and, as a result, different algorithms), we prove the closed-form expressions of the parameters $A, B, C, D_1, D_2 \geq 0, \rho \in (0, 1]$ for which Assumption 2.2 is satisfied.

## 3 GENERAL FRAMEWORK: PROXSKIP-VIP

Here we provide a unified algorithm framework (Algorithm 1) for solving regularized VIPs (2). Note that in this section, the guarantees hold under the general assumptions 2.1 and 2.2. In Section 4 we will further specify our results to the consensus reformulation setting where (3) also holds.

Algorithm 1 is inspired by the ProxSkip proposed in Mishchenko et al. (2022) for solving composite minimization problems. The two key elements of the algorithm are the randomized prox-skipping, and the control variate $h_t$. Via prox-skipping, the proximal oracle is rarely called if $p$ is small, which helps to reduce the computational cost when the proximal oracle is expensive. Also, for general regularized VIPs we have $F(x^*) \neq 0$ at the optimal point $x^*$ due to the composite structure. Thus skipping the proximal operator will not allow the method to converge. On this end, the introduction of $h_t$ alleviates such drift and stabilizes the iterations toward the optimal point ($h_t \to F(x^*)$).

We also highlight that Algorithm 1 is a general update rule and can vary based on the selection of the unbiased estimator $g_t$ and the probability $p$. For example, if $p = 1$, and $h_t \equiv 0$, the algorithm is reduced to the Proximal SGDA algorithm proposed in Beznosikov et al. (2022b) (with the associated theory). We will focus on further important special cases of Algorithm 1 below.

---

**Algorithm 1** ProxSkip-VIP

**Input:** Initial point $x_0$, parameters $\gamma, p$, initial control variate $h_0$, number of iterations $T$
 1: **for all** $t = 0, 1, ..., T$ **do**
 2:     $\widehat{x}_{t+1} = x_t - \gamma(g_t - h_t)$
 3:     Flip a coin $\theta_t$, and $\theta_t = 1$ w.p. $p$, otherwise 0
 4:     **if** $\theta_t = 1$ **then**
 5:         $x_{t+1} = \mathbf{prox}_{\frac{\gamma}{p}R}\left(\widehat{x}_{t+1} - \frac{\gamma}{p}h_t\right)$
 6:     **else**
 7:         $x_{t+1} = \widehat{x}_{t+1}$
 8:     **end if**
 9:     $h_{t+1} = h_t + \frac{p}{\gamma}(x_{t+1} - \widehat{x}_{t+1})$
10: **end for**

---

### 3.1 CONVERGENCE OF PROXSKIP-VIP

Our main convergence guarantees are presented in the following Theorem and Corollary.

**Theorem 3.1** (Convergence of ProxSkip-VIP). With Assumptions 2.1 and 2.2, let $\gamma \leq \min\left\{\frac{1}{\mu}, \frac{1}{2(A+MC)}\right\}$, $\tau \triangleq \min\left\{\gamma\mu, p^2, \rho - \frac{B}{M}\right\}$, for some $M > \frac{B}{\rho}$. Denote $V_t \triangleq \|x_t - x^*\|^2 + (\gamma/p)^2\|h_t - F(x^*)\|^2 + M\gamma^2\sigma_t^2$, Then the iterates of ProxSkip-VIP (Algorithm 1), satisfy:

$$\mathbb{E}[V_T] \leq (1 - \tau)^T V_0 + \frac{\gamma^2(D_1 + MD_2)}{\tau}.$$

Theorem 3.1 show that ProxSkip-VIP converges linearly to the neighborhood of the solution. The neighborhood is proportional to the step-size $\gamma$ and the parameters $D_1$ and $D_2$ of Assumption 2.2. In addition, we highlight that if we set $p = 1$, and $h_t \equiv 0$, then Theorem 3.1 recovers the convergence guarantees of Proximal-SGDA of (Beznosikov et al., 2022b, Theorem 2.2). As a corollary of Theorem 3.1, we can also obtain the following corresponding complexity results.

**Corollary 3.2.** With the setting in Theorem 3.1, if we set $M = \frac{2B}{\rho}$, $p = \sqrt{\gamma\mu}$ and $\gamma \leq$ $\min\left\{\frac{1}{\mu}, \frac{1}{2(A+MC)}, \frac{\rho}{2\mu}, \frac{\mu\epsilon}{2\left(D_1 + \frac{2B}{\rho}D_2\right)}\right\}$, we have $\mathbb{E}[V_T] \leq \epsilon$ with iteration complexity and the number of calls to the proximal oracle **prox**$(\cdot)$ as

$$\mathcal{O}\left(\max\left\{\frac{A + \frac{BC}{\rho}}{\mu}, \frac{1}{\rho}, \frac{D_1 + \frac{B}{\rho}D_2}{\mu^2\epsilon}\right\}\ln\frac{V_0}{\epsilon}\right) \quad \text{and} \quad \mathcal{O}\left(\sqrt{\max\left\{\frac{A + \frac{BC}{\rho}}{\mu}, \frac{1}{\rho}, \frac{D_1 + \frac{B}{\rho}D_2}{\mu^2\epsilon}\right\}}\ln\frac{V_0}{\epsilon}\right).$$

## 3.2 SPECIAL CASES OF GENERAL ANALYSIS

Theorem 3.1 holds under the general key Assumption 2.2 on the stochastic estimates. In this subsection, via Theorem 3.1, we explain how different selections of the unbiased estimator $g_t$ in Algorithm 1 lead to various convergence guarantees. In particular, here we cover (i) ProxSkip-SGDA, (ii) ProxSkip-GDA, and (iii)variance-reduced method ProxSkip-L-SVRGDA. To the best of our knowledge, none of these algorithms have been proposed and analyzed before for solving VIPs.

**(i) Algorithm: ProxSkip-SGDA.** Let us have the following assumption:

**Assumption 3.3** (Expected Cocoercivity). We assume that for all $t \geq 0$, the stochastic operator $g_t \triangleq g(x_t)$, which is an unbiased estimator of $F(x_t)$, satisfies expected cocoercivity, i.e., for all $x \in \mathbb{R}^d$ there is $L_g > 0$ such that $\mathbb{E}\|g(x) - g(x^*)\|^2 \leq L_g\langle F(x) - F(x^*), x - x^*\rangle$.

The expected cocoercivity condition was first proposed in Loizou et al. (2021) to analyze SGDA and the stochastic consensus optimization algorithms efficiently. It is strictly weaker compared to the bounded variance assumption and "growth conditions," and it implies the star-cocoercivity of the operator $F$. Assuming expected co-coercivity allows us to characterize the estimator $g_t$.

**Lemma 3.4** (Beznosikov et al. (2022b)). Let Assumptions 2.1 and 3.3 hold and let $\sigma_*^2 \triangleq \mathbb{E}\left[\|g(x^*) - F(x^*)\|^2\right] < +\infty$. Then $g_t$ satisfies Assumption 2.2 with $A = L_g$, $D_1 = 2\sigma_*^2$, $\rho = 1$ and $B = C = D_2 = \sigma_t^2 \equiv 0$.

By combining Lemma 3.4 and Corollary 3.2, we obtain the following result.

**Corollary 3.5** (Convergence of ProxSkip-SGDA). With Assumption 2.1 and 3.3, if we further set $\gamma \leq \min\left\{\frac{1}{2L_g}, \frac{1}{2\mu}, \frac{\mu\epsilon}{8\sigma_*^2}\right\}$ and $p = \sqrt{\gamma\mu}$, then for the iterates of ProxSkip-SGDA, we have $\mathbb{E}[V_T] \leq \epsilon$ with iteration complexity and the number of calls of the proximal oracle **prox**$(\cdot)$ as $\mathcal{O}\left(\max\left\{\frac{L_g}{\mu}, \frac{\sigma_*^2}{\mu^2\epsilon}\right\}\ln\frac{1}{\epsilon}\right)$ and $\mathcal{O}\left(\sqrt{\max\left\{\frac{L_g}{\mu}, \frac{\sigma_*^2}{\mu^2\epsilon}\right\}}\ln\frac{1}{\epsilon}\right)$, respectively.

**(ii) Deterministic Case: ProxSkip-GDA.** In the deterministic case where $g(x_t) = F(x_t)$, the algorithm ProxSkip-SGDA is reduced to ProxSkip-GDA. Thus, Lemma 3.4 and Corollary 3.2 but with $\sigma_*^2 = 0$. In addition, the expected co-coercivity parameter becomes $L_g = \ell$ by Assum. 2.1.

**Corollary 3.6** (Convergence of ProxSkip-GDA). With the same setting in Corollary 3.5, if we set the estimator $g_t = F(x_t)$, and $\gamma = \frac{1}{2\ell}$ the iterates of ProxSkip-GDA satisfy: $\mathbb{E}[V_T] \leq \left(1 - \min\left\{\gamma\mu, p^2\right\}\right)^T V_0$, and we get $\mathbb{E}[V_T] \leq \epsilon$ with iteration complexity and number of calls of the proximal oracle **prox**$(\cdot)$ as $\mathcal{O}\left(\frac{\ell}{\mu}\ln\frac{1}{\epsilon}\right)$ and $\mathcal{O}\left(\sqrt{\frac{\ell}{\mu}}\ln\frac{1}{\epsilon}\right)$, respectively.

Note that if we further have $F$ to be $L$-Lipschitz continuous and $\mu$-strongly monotone, then $\ell = \kappa L$ where $\kappa = L/\mu$ is the condition number (Loizou et al., 2021). Thus the number of iteration and calls of the proximal oracles are $\mathcal{O}\left(\kappa^2\ln\frac{1}{\epsilon}\right)$ and $\mathcal{O}\left(\kappa\ln\frac{1}{\epsilon}\right)$ respectively. In the minimization setting, these two complexities are equal to $\mathcal{O}\left(\kappa\ln\frac{1}{\epsilon}\right)$ and $\mathcal{O}\left(\sqrt{\kappa}\ln\frac{1}{\epsilon}\right)$ since $L_g = \ell = L$. In this case, our result recovers the result of the original ProxSkip method in Mishchenko et al. (2022).

**(iii) Algorithm: ProxSkip-L-SVRGDA.** Here, we focus on a variance-reduced variant of the proposed ProxSkip framework (Algorithm 1). We further specify the operator $F$ in (2) to be in a finite-

sum formulation: $F(x) = \frac{1}{n} \sum_{i=1}^{n} F_i(x)$. We propose the ProxSkip-Loopless-SVRG (ProxSkip-L-SVRGDA) algorithm (Algorithm 3 in Appendix) which generalizes the L-SVRGDA proposed in Beznosikov et al. (2022b). In this setting, we need to introduce the following assumption:

**Assumption 3.7.** We assume that there exist a constant $\widehat{\ell}$ such that for all $x \in \mathbb{R}^d$: $\frac{1}{n} \sum_{i=1}^{n} \|F_i(x) - F_i(x^*)\|^2 \le \widehat{\ell}\langle F(x) - F(x^*), x - x^* \rangle$.

If each $F_i$ is $\ell_i$-cocoercive, then Assumption 3.7 holds with $\widehat{\ell} \le \max_{i \in [n]} L_i$. Using Assumption 3.7 we obtain the following result (Beznosikov et al., 2022b).

**Lemma 3.8** (Beznosikov et al. (2022b)). With Assumption 2.1 and 3.7, we have the estimator $g_t$ in Algorithm 3 satisfies Assumption 2.1 with $A = \widehat{\ell}$, $B = 2$, $D_1 = D_2 = 0$, $C = \frac{q\widehat{\ell}}{2}$, $\rho = q$ and $\sigma_t^2 = \frac{1}{n} \sum_{i=1}^{n} \|F_i(x_t) - F_i(x^*)\|^2$.

By combining Lemma 3.8 and Corollary 3.2, we obtain the following result for ProxSkip-L-SVRGDA:

**Corollary 3.9** (Complexities of ProxSkip-L-SVRGDA). Let Assumption 2.1 and 3.7 hold. If we further set $q = 2\gamma\mu$, $M = \frac{4}{q}$, $p = \sqrt{\gamma\mu}$ and $\gamma = \min\left\{\frac{1}{\mu}, \frac{1}{6\widehat{\ell}}\right\}$, then we obtain $\mathbb{E}[V_T] \le \epsilon$ with iteration complexity and the number of calls of the proximal oracle $\mathbf{prox}(\cdot)$ as $\mathcal{O}(\widehat{\ell}/\mu \ln \frac{1}{\epsilon})$ and the number of calls of the proximal oracle $\mathbf{prox}(\cdot)$ as $\mathcal{O}(\sqrt{\widehat{\ell}/\mu} \ln \frac{1}{\epsilon})$.

If we further consider Corollary 3.9 in the minimization setting and assume each $F_i$ is $L$-Lipschitz and $\mu$-strongly convex, the number of iteration and calls of the proximal oracles will be $\mathcal{O}(\kappa \ln \frac{1}{\epsilon})$ and $\mathcal{O}(\sqrt{\kappa} \ln \frac{1}{\epsilon})$, which recover the results of variance-reduced ProxSkip in Malinovsky et al. (2022).

## 4 APPLICATION OF PROXSKIP TO FEDERATED LEARNING

In this section, by specifying the general problem to the expression of (3) as a special case of (2), we explain how the proposed algorithmic framework can be interpreted as federated learning algorithms. As discussed in Section 2.1, in the distributed/federated setting, evaluating the $\mathbf{prox}_{\gamma R}(x)$ is equivalent to a communication between the $n$ workers. In the FL setting, Algorithm 1 can be expressed as Algorithm 2. Note that skipping the proximal operator in Algorithm 1 corresponds to local updates in Algorithm 2. In Algorithm 2, $g_{i,t} = g_i(x_{i,t})$ is the unbiased estimator of the $f_i(x_{i,t})$ of the original problem (1), while the client control vectors $h_{i,t}$ satisfy $h_{i,t} \to f_i(z^*)$. The probability $p$ in this setting shows how often a communication takes place (averaging of the workers' models).

---

**Algorithm 2** ProxSkip-VIP-FL

**Input:** Initial points $\{x_{i,0}\}_{i=1}^{n}$ and $\{h_{i,0}\}_{i=1}^{n}$, parameters $\gamma, p, T$
1: **for all** $t = 0, 1, ..., T$ **do**
2:  **Server:** Flip a coin $\theta_t$, $\theta_t = 1$ w.p. $p$, otherwise 0. Send $\theta_t$ to all workers
3:  **for each workers** $i \in [n]$ **in parallel do**
4:   $\widehat{x}_{i,t+1} = x_{i,t} - \gamma(g_{i,t} - h_{i,t})$                              // Local update with control variate
5:   **if** $\theta_t = 1$ **then**
6:    Worker: $x'_{i,t+1} = \widehat{x}_{i,t+1} - \frac{\gamma}{p} h_{i,t}$, sends $x'_{i,t+1}$ to the server
7:    Server: computes $x_{i,t+1} = \frac{1}{n} \sum_{i=1}^{n} x'_{i,t+1}$ and send to workers          // Communication
8:   **else**
9:    $x_{i,t+1} = \widehat{x}_{i,t+1}$                              // Otherwise skip the communication step
10:   **end if**
11:   $h_{i,t+1} = h_{i,t} + \frac{p}{\gamma}(x_{i,t+1} - \widehat{x}_{i,t+1})$
12:  **end for**
13: **end for**

---

**Algorithm: ProxSkip-SGDA-FL.** The first implementation of the framework we consider is ProxSkip-SGDA-FL. Similar to ProxSkip-SGDA in the centralized setting, here we set the estimator to be the vanilla estimator of $F$, i.e., $g_{i,t} = g_i(x_{i,t})$, where $g_i$ is an unbiased estimator of $f_i$.

We note that if we set $h_{i,t} \equiv 0$, then Algorithm 2 is reduced to the typical Local SGDA (Deng & Mahdavi, 2021). To proceed with the analysis in FL, we require the following assumption:

> **Assumption 4.1.** The Problem (1) attains a unique solution $z^* \in \mathbb{R}^{d'}$. Each $f_i$ in (1), it is $\mu$-quasi-strongly monotone around $z^*$, i.e., for any $x_i \in \mathbb{R}^{d'}$, $\langle f_i(x_i) - f_i(z^*), x_i - z^* \rangle \geq \mu \|x_i - z^*\|^2$. Operator $g_i(x_i)$, is an unbiased estimator of $f_i(x_i)$, and for all $x_i \in \mathbb{R}^{d'}$ we have $\mathbb{E}\|g_i(x_i) - g_i(z^*)\|^2 \leq L_g \langle f_i(x_i) - f_i(z^*), x_i - z^* \rangle$.

The assumption on the uniqueness of the solution is pretty common in the literature. For example, in the (unconstrained) minimization case, quasi-strong monotonicity implies uniqueness $z^*$ (Hinder et al., 2020). Assumption 4.1 is required as through it, we can prove that the operator $F$ in (2) satisfies Assumption 2.1, and its corresponding estimator satisfies Assumption 2.2, which we detail in Appendix F.3. Let us now present the convergence guarantees.

> **Theorem 4.2** (Convergence of ProxSkip-SGDA-FL). With Assumption 4.1, then ProxSkip-VIP-FL (Algorithm 2) achieves $\mathbb{E}[V_T] \leq \epsilon$ (where $V_T$ is defined in Theorem 3.1), with iteration complexity $\mathcal{O}(\max\left\{\frac{L_g}{\mu}, \frac{\sigma_*^2}{\mu^2 \epsilon}\right\} \ln \frac{1}{\epsilon})$ and communication complexity $\mathcal{O}(\sqrt{\max\left\{\frac{L_g}{\mu}, \frac{\sigma_*^2}{\mu^2 \epsilon}\right\}} \ln \frac{1}{\epsilon})$.

**Comparison with Literature.** Note that Theorem 4.2 is quite general, which holds under any reasonable, unbiased estimator. In the special case of federated minimax problems, one can use the same (mini-batch) gradient estimator from Local SGDA (Deng & Mahdavi, 2021) in Algorithm 2 and our results still hold. The benefit of our approach compared to Local SGDA is the communication acceleration, as pointed out in Table 1. In addition, in the deterministic setting ($g_i(x_{i,t}) = f_i(x_{i,t})$) we have $\sigma_*^2 = 0$ and Theorem 4.2 reveals $\mathcal{O}(\ell/\mu \ln \frac{1}{\epsilon})$ iteration complexity and $\mathcal{O}(\sqrt{\ell/\mu} \ln \frac{1}{\epsilon})$ communication complexity for ProxSkip-GDA-FL. In Table 2 of the appendix, we provide a more detailed comparison of our Algorithm 2 (Theorem 4.2) with existing literature in FL. The proposed approach outperforms other algorithms (Local SGDA, Local SEG, FedAvg-S) in terms of iteration and communication complexities.

As the baseline, the distributed (centralized) gradient descent-ascent (GDA) and extragradient (EG) algorithms achieve $\mathcal{O}(\kappa^2 \ln \frac{1}{\epsilon})$ and $\mathcal{O}(\kappa \ln \frac{1}{\epsilon})$ communication complexities, respectively (Fallah et al., 2020; Mokhtari et al., 2020b). We highlight that our analysis does not require an assumption on bounded heterogeneity / dissimilarity, and as a result, we can solve problems with heterogeneous data. Finally, as reported by Beznosikov et al. (2020), the lower communication complexity bound for problem (1) is given by $\Omega(\kappa \ln \frac{1}{\epsilon})$, which further highlights the optimality of our proposed ProxSkip-SGDA-FL algorithm.

**Algorithm: ProxSkip-L-SVRGDA-FL.** Next, we focus on variance-reduced variants of ProxSkip-SGDA-FL and we further specify the operator $F$ in (2) as $F(x) \triangleq \sum_{i=1}^{n} F_i(x_i)$ and $F_i(x) \triangleq \frac{1}{m_i} \sum_{j=1}^{m_i} F_{i,j}(x_i)$. The proposed algorithm, ProxSkip-L-SVRGDA-FL, is presented in the Appendix as Algorithm 4. In this setting, we need the following assumption on $F_i$ to proceed with the analysis.

> **Assumption 4.3.** The Problem (1) attains a unique solution $z^*$. Also for each $f_i$ in (1), it is $\mu$-quasi-strongly monotone around $z^*$. Moreover we assume for all $x_i \in \mathbb{R}^{d'}$ we have $\frac{1}{m_i} \sum_{j=1}^{m_i} \|F_{i,j}(x_i) - F_{i,j}(z^*)\|^2 \leq \widehat{\ell} \langle f_i(x_i) - f_i(z^*), x_i - z^* \rangle$.

Similar to the ProxSkip-SGDA-FL case, we can show that under Assumption (4.3), the operator and unbiased estimator fit into the setting of Assumptions 2.1 and 2.2 (see derivation in Appendix F.4). As a result, we can obtain the following complexity result.

> **Theorem 4.4** (Convergence of ProxSkip-L-SVRGDA-FL). Let Assumption 4.3 hold. Then the iterates of ProxSkip-L-SVRGDA-FL achieve $\mathbb{E}[V_T] \leq \epsilon$ (where $V_T$ is defined in Theorem 3.1) with iteration complexity $\mathcal{O}(\widehat{\ell}/\mu \ln \frac{1}{\epsilon})$ and communication complexity $\mathcal{O}(\sqrt{\widehat{\ell}/\mu} \ln \frac{1}{\epsilon})$.

In the special case of minimization problems $\min_{z \in \mathbb{R}^d} f(z)$, i.e., $F(z) = \nabla f(z)$, we have Theorem 4.4 recovers the theoretical result of Malinovsky et al. (2022) which focuses on ProxSkip methods for minimization problems, showing the tightness of our approach. Similar to the ProxSkip-VIP-FL, the ProxSkip-L-SVRGDA-FL enjoys a communication complexity improvement in terms of the condition number $\kappa$. For more details, please refer to Table 1 (and Table 2 in the appendix).

## 5 NUMERICAL EXPERIMENTS

We corroborate our theory with the experiments, and test the performance of the proposed algorithms ProxSkip-(S)GDA-FL (Algorithm 2 with $g_i(x_{i,t}) = f_i(x_{i,t})$ or its unbiased estimator) and ProxSkip-L-SVRGDA-FL (Algorithm 4). We focus on two classes of problems: (i) strongly monotone quadratic games and (ii) robust least squares. See Appendix G and H for more details and extra experiments.

To evaluate the performance, we use the relative error measure $\frac{\|x_k - x^*\|^2}{\|x_0 - x^*\|^2}$. The horizontal axis corresponds to the number of communication rounds. For all experiments, we pick the step-size $\gamma$ and probability $p$ for different algorithms according to our theory. That is, ProxSkip-(S)GDA-FL based on Corollary 3.5 and 3.6, ProxSkip-L-SVRGDA-FL by Corollary 3.9. See also Table 3 in the appendix for the settings of parameters. We compare our methods to Local SGDA (Deng & Mahdavi, 2021), Local SEG (Beznosikov et al., 2020) (and their deterministic variants), and FedGDA-GT, a deterministic algorithm proposed in Sun & Wei (2022). For all methods, we use parameters based on their theoretical convergence guarantees. For more details on our implementations and additional experiments, see Appendix G and H.

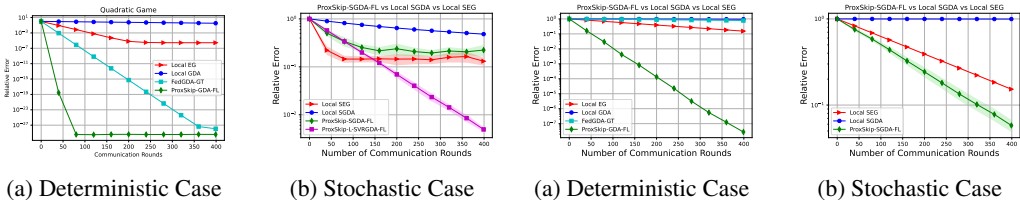

| (a) Deterministic Case | (b) Stochastic Case | (a) Deterministic Case | (b) Stochastic Case |

Figure 1: Comparison of algorithms on the strongly-monotone quadratic game (5).

Figure 2: Comparison of algorithms on the Robust Least Square (6)

**Strongly-monotone Quadratic Games.** In the first experiment, we consider a problem of the form

$$\min_{x_1 \in \mathbb{R}^d} \max_{x_2 \in \mathbb{R}^d} \frac{1}{n} \sum_{i=1}^{n} \frac{1}{m_i} \sum_{j=1}^{m_i} f_{ij}(x_1, x_2), \tag{5}$$

where $f_{ij}(x_1, x_2) \triangleq \frac{1}{2} x_1^\mathsf{T} A_{ij} x_1 + x_1^\mathsf{T} B_{ij} x_2 - \frac{1}{2} x_1^\mathsf{T} C_{ij} x_1 + a_{ij}^\mathsf{T} x_1 - c_{ij}^\mathsf{T} x_2$. Here we set the number of clients $n = 20$, $m_i = 100$ for all $i \in [n]$ and $d = 20$. We generate positive semidefinite matrices $A_{ij}, B_{ij}, C_{ij} \in \mathbb{R}^{d \times d}$ such that eigenvalues of $A_{ij}, C_{ij}$ lie in the interval $[0.01, 1]$ while those of $B_{ij}$ lie in $[0, 1]$. The vectors $a_{ij}, c_{ij} \in \mathbb{R}^d$ are generated from $\mathcal{N}_d(0, I_d)$ distribution. This data generation process ensures that the quadratic game satisfies the assumptions of our theory. To make the data heterogeneous, we produce different $A_{ij}, B_{ij}, C_{ij}, a_{ij}, c_{ij}$ across the clients indexed by $i \in [n]$.

We present the results in Figure 1 for both deterministic and stochastic settings. As our theory predicted, our proposed methods are always faster in terms of communication rounds than Local (S)GDA. The current analysis of Local EG requires the bounded heterogeneity assumption which leads to convergence to a neighborhood even in a deterministic setting. As also expected by the theory, our proposed variance-reduced algorithm converges linearly to the exact solution.

**Robust Least Square.** In the second experiment, we consider the robust least square (RLS) problem (El Ghaoui & Lebret, 1997; Yang et al., 2020a) with the coefficient matrix $\mathbf{A} \in \mathbb{R}^{r \times s}$ and noisy vector $y_0 \in \mathbb{R}^r$. We assume $y_0$ is corrupted by a bounded perturbation $\delta$ (i.e., $\|\delta\| \leq \delta_0$). RLS minimizes the worst case residual and can be formulated as follows:

$$\min_{\beta} \max_{\delta:\|\delta\| \leq \delta_0} \|\mathbf{A}\beta - y\|^2 \quad \text{where} \quad \delta = y_0 - y.$$

In our work, we consider the following penalized version of the RLS problem:

$$\min_{\beta \in \mathbb{R}^s} \max_{y \in \mathbb{R}^r} \|\mathbf{A}\beta - y\|^2 - \lambda \|y - y_0\|^2 \tag{6}$$

This objective function is strongly-convex-strongly-concave when $\lambda > 1$ (Thekumparampil et al., 2022). We run our experiment on the "California Housing" dataset from scikit-learn package (Pedregosa et al., 2011), using $\lambda = 50$ in (6). Please see Appendix G for further detail on the setting.

In Figure 2, we show the trajectories of our proposed algorithms compared to Local EG, Local GDA and their stochastic variants. In both scenarios (deterministic and stochastic), our ProxSkip-GDA/SGDA-FL algorithms outperform the other methods in terms of communication rounds.

## REPRODUCIBILITY STATEMENT

The code for implementing all proposed algorithms and reproducing our experimental evaluation is available within the supplementary materials, and at https://github.com/isayantan/ProxSkipVIP.

## ACKNOWLEDGEMENT

Siqi Zhang gratefully acknowledges funding support from The Acheson J. Duncan Fund for the Advancement of Research in Statistics by Johns Hopkins University. Nicolas Loizou acknowledges support from CISCO Research.

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

# Supplementary Material

The Supplementary Material is organized as follows:

Section A reviews the existing literature related to our work. Section B summarizes the notational conventions used in the main paper and appendix, while Section C includes the missing pseudocodes of the main paper (related to variance reduced methods). Next, Sections D and E present some basic lemmas to prove our main convergence results. We present the proofs of the main theorems and corollaries in Section F. Finally, in section G, we add further details related to our experiments in the main paper, while in section H, we add more related experiments to verify our theory.

## CONTENTS

## A   FURTHER RELATED WORK

The references necessary to motivate our work and connect it to the most relevant literature are included in the appropriate sections of the main body of the paper. In this section, we present a broader view of the literature, including more details on closely related work and more references to papers that are not directly related to our main results.

**Federated Learning.**   One of the most popular characteristics of communication-efficient FL algorithms is local updates, where each client/node of the network takes multiple steps of the chosen optimization algorithm locally between the communication rounds. Many algorithms with local updates (e.g., FedAvg, Local GD, Local SGD, etc.) have been proposed and extensively analyzed in FL literature under various settings, including convex and nonconvex problems (McMahan et al., 2017; Stich, 2018; Assran et al., 2019; Kairouz et al., 2021; Wang et al., 2021; Karimireddy et al., 2020; Woodworth et al., 2020; Koloskova et al., 2020). In local FL algorithms, a big source of variance in the convergence guarantees stems from *inter-client* differences (heterogeneous data). On that end, inspired by the popular variance reduction methods from optimization literature (e.g., SVRG (Johnson & Zhang, 2013) or SAGA (Defazio et al., 2014)), researchers start investigating variance reduction mechanisms to handle the variance related to heterogeneous environments (e.g. FedSVRG (Konečný et al., 2016)), resulting in the popular SCAFFOLD algorithm(Karimireddy et al., 2020). Gradient tracking (Di Lorenzo & Scutari, 2016; Nedic et al., 2017) is another class of methods not affected by data-heterogeneity, but its provable communication complexity scales linearly in the condition number (Koloskova et al., 2021; Alghunaim & Yuan, 2023), even when combined with local steps (Liu et al., 2023). Finally, Mishchenko et al. (2022) proposed a new algorithm (ProxSkip) for federated minimization problems that guarantees acceleration of communication complexity under heterogeneous data. As we mentioned in Section 1.1, the framework of our paper includes the algorithm and convergence results of Mishchenko et al. (2022) as a special case, and it could be seen as an extension of these ideas in the distributed variational inequality setting.

**Minimax Optimization and Variational Inequality Problems.**   Minimax optimization, and more generally, variational inequality problems (VIPs) (Hartman & Stampacchia, 1966; Facchinei & Pang, 2003) appear in various research areas, including but not limited to online learning (Cesa-Bianchi & Lugosi, 2006), game theory (Von Neumann & Morgenstern, 1947), machine learning (Goodfellow et al., 2014) and social and economic practices (Facchinei & Pang, 2003; Parise & Ozdaglar, 2019). The gradient descent-ascent (GDA) method is one of the most popular algorithms for solving minimax problems. However, GDA fails to converge for convex-concave minimax problems (Mescheder et al., 2018) or bilinear games (Gidel et al., 2019). To avoid these issues, Korpelevich (1976) introduced the extragradient method (EG), while Popov (1980) proposed the optimistic gradient method (OG). Recently, there has been a surge of developments for improving the extra gradient methods for better convergence guarantees. For example, Mokhtari et al. (2020b;a) studied optimistic and extragradient methods as an approximation of the proximal point method to solve bilinear games, while Hsieh et al. (2020) focused on the stochastic setting and proposed the use of a larger extrapolation step size compared to update step size to prove convergence under error-bound conditions.

Beyond the convex-concave setting, most machine learning applications expressed as min-max optimization problems involve nonconvex-nonconcave objective functions. Recently, Daskalakis et al. (2021) showed that finding local solutions is intractable in the nonconvex-nonconcave regime,

which motivates researchers to look for additional structures on problems that can be exploited to prove the convergence of the algorithms. For example, Loizou et al. (2021) provided convergence guarantees of stochastic GDA under expected co-coercivity, while Yang et al. (2020a) studied the convergence stochastic alternating GDA under the PL-PL condition. Lin et al. (2020) showed that solving nonconvex-(strongly)-concave minimax problems using GDA by appropriately maintaining the primal and dual stepsize ratio is possible, and several follow-up works improved the computation complexities (Yang et al., 2020b; Zhang et al., 2021b; Yang et al., 2022b; Zhang et al., 2022). Later Diakonikolas et al. (2021) introduced the notion of the weak minty variational inequality (MVI), which captures a large class of nonconvex-nonconcave minimax problems. Gorbunov et al. (2022b) provided tight convergence guarantees for solving weak MVI using deterministic extragradient and optimistic gradient methods. However, deterministic algorithms can be computationally expensive, encouraging researchers to look for stochastic algorithms. On this end, Diakonikolas et al. (2021); Pethick et al. (2023a); Böhm (2022) analyze stochastic extragradient and optimistic gradient methods for solving weak MVI with increasing batch sizes. Recently, Pethick et al. (2023b) used a bias-corrected variant of the extragradient method to solve weak MVI without increasing batch sizes. Lately, variance reduction methods has also proposed for solving min-max optimization problems and VIPs Alacaoglu & Malitsky (2022); Cai et al. (2022). As we mentioned in the main paper, in our work, the proposed convergence guarantees hold for a class of non-monotone problems, i.e., the class of is $\mu$-quasi-strongly monotone and $\ell$-star-cocoercive operators (see Assumption 2.1).

**Minimax Federated Learning.** Going beyond the more classical centralized setting, recent works study the two-player federated minimax problems (Deng & Mahdavi, 2021; Beznosikov et al., 2020; Sun & Wei, 2022; Hou et al., 2021; Sharma et al., 2022; Tarzanagh et al., 2022; Huang, 2022; Yang et al., 2022a). For example, Deng & Mahdavi (2021) studied the convergence guarantees of Local SGDA, which is an extension of Local SGD in the minimax setting, under various assumptions; Sharma et al. (2022) studied furthered and improved the complexity results of Local SGDA in the nonconvex case; Beznosikov et al. (2020) studied the convergence rate of Local SGD algorithm with an extra step (we call it Local Stochastic Extragradient or Local SEG), under the (strongly)-convex–(strongly)-concave setting. Multi-player games, which, as we mentioned in the main paper, can be formulated as a special case of the VIP, are well studied in the context of game theory (Rosen, 1965; Cai & Daskalakis, 2011; Yi & Pavel, 2017; Chen et al., 2023). With our work, by studying the more general distributed VIPs, our proposed federated learning algorithms can also solve multi-player games.

Finally, as we mentioned in the main paper, the algorithmic design of our methods is inspired by the proposed algorithm, ProxSkip of Mishchenko et al. (2022), for solving composite minimization problems. However, we should highlight that since Mishchenko et al. (2022) focuses on solving optimization problems, the function suboptimality $[f(x^k) - f(x^*)]$ is available (a concept that cannot be useful in the VIP setting. Thus, the difference in the analysis between the two papers begins at a deeper conceptual level. In addition, our work provides a unified algorithm framework under a general estimator setting (Assumption 2.2) in the VIP regime that captures variance-reduced GDA and the convergence guarantees from Beznosikov et al. (2022b) as a special case. Compared to existing works on federated minimax optimization (Beznosikov et al., 2020; Deng & Mahdavi, 2021), our analysis provides improved communication complexities and avoids the restrictive (uniform) bounded heterogeneity/variance assumptions (see discussion in Section 4).

# B    NOTATIONS

Here we present a summary of the most important notational conventions used throughout the paper.

- FL: Federated Learning
- VIP: Variational Inequality Problem
- $\ell$: Modulus of star-cocoercivity
- $\widehat{\ell}$: Modulus of averaged star-cocoercivity
- $L$: Modulus of Lipschitz continuity
- $L_g$: Modulus of expected cocoercivity
- $\mu$: Modulus of (quasi-)strong monotonicity
- $A, B, C, D_1, D_2 \geq 0$, $\rho \in (0,1]$: Parameters of Assumption 2.2 on the estimator $g$
- $\kappa = \frac{L}{\mu}$: condition number
- $[x]_{i:j}$: the $i$-th to $j$-th coordinates of vector $x$
- $[n]$: the set $\{1, 2, \cdots, n\}$
- $z \in \mathbb{R}^{d'}$: the variable in VIP (1), $F(z) = \frac{1}{n} \sum_{i=1}^{n} f_i(z)$, $f_i : \mathbb{R}^{d'} \to \mathbb{R}^{d'}$
- $x \in \mathbb{R}^d$: the variable in regularized VIP (2), $d = nd'$, $F(x) \triangleq \frac{1}{n} \sum_{i=1}^{n} F_i(x_i)$. $x = (x_1, x_2, \cdots, x_n) \in \mathbb{R}^d$, $x_i \in \mathbb{R}^{d'}$.
- $F : \mathbb{R}^d \to \mathbb{R}^d$, $F_i : \mathbb{R}^{d'} \to \mathbb{R}^d$ and $F_i(x_i) = (0, \cdots, f_i(x_i), \cdots, 0)$ where $[F(x)]_{(id'+1):(id'+d')} = f_i(x_i)$.
- $\sigma_*^2 \triangleq \mathbb{E}\left[\|g(x^*) - F(x^*)\|^2\right] < +\infty$
- $\Delta$: (uniform) bounded variance of operator estimator

## C FURTHER PSEUDOCODES

Here we present the pseudocodes of the algorithms that due to space limitation did not fit into the main paper.

We present the ProxSkip-L-SVRGDA method in Algorithm 3, which can be seen as a ProxSkip generalization of the L-SVRGDA algorithm proposed in Beznosikov et al. (2022b). For more details, please refer to Section 3.2. Note that the unbiased estimator in this method has the form: $g_t = F_{j_t}(x_t) - F_{j_t}(w_t) + F(w_t)$.

---

**Algorithm 3** ProxSkip-L-SVRGDA

---

**Input:** Initial point $x_0, h_0$, parameters $\gamma$, probabilities $p, q \in [0, 1]$, number of iterations $T$
1: Set $w_0 = x_0$, compute $F(w_0)$
2: **for all** $t = 0, 1, ..., T$ **do**
3:      Construct $g_t = F_{j_t}(x_t) - F_{j_t}(w_t) + F(w_t)$, $j_t \in [n]$
4:      Update $w_{t+1} = \begin{cases} x_t & \text{w.p. } q \\ w_t & \text{w.p. } 1 - q \end{cases}$
5:      $\widehat{x}_{t+1} = x_t - \gamma(g_t - h_t)$
6:      Flip a coin $\theta_t$, $\theta_t = 1$ w.p. $p$, otherwise 0
7:      **if** $\theta_t = 1$ **then**
8:          $x_{t+1} = \mathbf{prox}_{\frac{\gamma}{p} R}\left(\widehat{x}_{t+1} - \frac{\gamma}{p} h_t\right)$
9:      **else**
10:         $x_{t+1} = \widehat{x}_{t+1}$
11:      **end if**
12:      $h_{t+1} = h_t + \frac{p}{\gamma}(x_{t+1} - \widehat{x}_{t+1})$
13: **end for**
**Output:** $x_T$

---

Moreover, we present the ProxSkip-L-SVRGDA-FL algorithm below in Algorithm 4. Similar to the relationship between ProxSkip-SGDA and ProxSkip-SGDA-FL, here this algorithm is the implementation of the ProxSkip-L-SVRGDA method mentioned above in the FL regime. For more details, please check Section 4 of the main paper.

---

**Algorithm 4** ProxSkip-L-SVRGDA-FL

---

**Input:** Initial points $x_{1,0}, \cdots, x_{n,0} \in \mathbb{R}^{d'}$, $\gamma, p \in \mathbb{R}$, initial control variates $h_{1,0}, \cdots, h_{n,0} = 0 \in \mathbb{R}^{d'}$, iteration number $T$

1: Set $w_{i,0} = x_{i,0}$ for every worker $i \in [n]$
2: **for all** $t = 0, 1, ..., T$ **do**
3:     **Server:** Flip two coins $\theta_t$ and $\zeta_t$, where $\theta_t = 1$ w.p. $p$ and $\zeta_t = 1$ w.p. $q$, otherwise 0. Send $\theta_t$ and $\zeta_t$ to all workers
4:     **for each workers** $i \in [n]$ **in parallel do**
5:         $g_{i,t} = F_{i,j_t}(x_{i,t}) - F_{i,j_t}(w_{i,t}) + F_i(w_{i,t})$, where $j_t \sim \text{Unif}([m_i])$
6:         Update $w_{i,t+1} = \begin{cases} x_{i,t} & \text{if } \zeta_t = 1 \\ w_{i,t} & \text{otherwise} \end{cases}$
7:         $\widehat{x}_{i,t+1} = x_{i,t} - \gamma(g_{i,t} - h_{i,t})$
8:         **if** $\theta_t = 1$ **then**
9:             Worker: $x'_{i,t+1} = \widehat{x}_{i,t+1} - \frac{\gamma}{p}h_{i,t}$, sends $x'_{i,t+1}$ to the server
10:           Server: computes $x_{i,t+1} = \frac{1}{n}\sum_{i=1}^{n} x'_{i,t+1}$, sends $x_{i,t+1}$ to workers     // Communication
11:         **else**
12:            $x_{i,t+1} = \widehat{x}_{i,t+1}$                 // Otherwise skip the communication step
13:         **end if**
14:         $h_{i,t+1} = h_{i,t} + \frac{p}{\gamma}(x_{i,t+1} - \widehat{x}_{i,t+1})$
15:     **end for**
16: **end for**
**Output:** $x_T$

---

## D   USEFUL LEMMAS

**Lemma D.1** (Young's Inequality).

$$\|a + b\|^2 \leq 2\|a\|^2 + 2\|b\|^2 \tag{7}$$

**Lemma D.2.** For any optimal solution $x^*$ of (2) and any $\alpha > 0$, we have

$$x^* = \mathbf{prox}_{\alpha R}(x^* - \alpha F(x^*)). \tag{8}$$

**Lemma D.3** (Firm Nonexpansivity of the Proximal Operator (Beck, 2017)). Let $f$ be a proper closed and convex function, then for any $x, y \in \mathbb{R}^d$ we have

$$\langle x - y, \mathbf{prox}_f(x) - \mathbf{prox}_f(y)\rangle \geq \left\|\mathbf{prox}_f(x) - \mathbf{prox}_f(y)\right\|^2, \tag{9}$$

or equivalently,

$$\left\|\left(x - \mathbf{prox}_f(x)\right) - \left(y - \mathbf{prox}_f(y)\right)\right\|^2 + \left\|\mathbf{prox}_f(x) - \mathbf{prox}_f(y)\right\|^2 \leq \|x - y\|^2. \tag{10}$$

# E    PROOFS OF LEMMAS 3.4 AND 3.8

As we mentioned in the main paper, the Lemmas 3.4 and 3.8 have been proved in Beznosikov et al. (2022b). We include the proofs of these results using our notation for completeness.

*Proof of Lemma 3.4.*  Note that

$$\mathbb{E}\left\|g_t - F(x^*)\right\|^2 \overset{(7)}{\leq} 2\,\mathbb{E}\left\|g_t - g(x^*)\right\|^2 + 2\,\mathbb{E}\left\|g(x^*) - F(x^*)\right\|^2$$
$$\leq 2L_g\langle F(x_t) - F(x^*), x_t - x^*\rangle + 2\sigma_*^2,$$

where the second inequality uses Assumption 3.3. The statement of Lemma 3.4 is obtained by comparing the above inequality with the expression of Assumption 2.2.  □

*Proof of Lemma 3.8.*  For the property of $g_t$, it is easy to find that the unbiasedness holds. Then note that

$$\mathbb{E}\|g_t - F(x^*)\|^2$$
$$= \mathbb{E}\|F_{j_t}(x_t) - F_{j_t}(w_t) + F(w_{j_t}) - F(x^*)\|^2$$
$$= \frac{1}{n}\sum_{i=1}^{n}\|F_i(x_t) - F_{j_t}(w_t) + F(w_{j_t}) - F(x^*)\|^2$$
$$\overset{(7)}{\leq} \frac{2}{n}\sum_{i=1}^{n}\|F_i(x_t) - F_i(x^*)\|^2 + \|F_i(x^*) - F_i(w_t) - (F(x^*) - F(w_t))\|^2$$
$$\leq \frac{2}{n}\sum_{i=1}^{n}\|F_i(x_t) - F_i(x^*)\|^2 + \|F_i(x^*) - F_i(w_t)\|^2$$
$$\leq 2\widehat{\ell}\langle F(x_t) - F(x^*), x_t - x^*\rangle + \frac{2}{n}\sum_{i=1}^{n}\|F_i(x^*) - F_i(w_t)\|^2$$
$$= 2\widehat{\ell}\langle F(x_t) - F(x^*), x_t - x^*\rangle + 2\sigma_t^2,$$

here the second inequality comes from the fact that $\mathrm{Var}(X) \leq \mathbb{E}(X^2)$, and the third inequality comes from Assumption 3.7. Then for the second term above, we have

$$\mathbb{E}\left[\sigma_{t+1}^2\right] = \frac{1}{n}\sum_{i=1}^{n}\|F_i(w_{t+1}) - F_i(x^*)\|^2$$
$$= \frac{1}{n}\sum_{i=1}^{n}\left(q\|F_i(x_t) - F_i(x^*)\|^2 + (1-q)\|F_i(w_t) - F_i(x^*)\|^2\right)$$
$$\leq q\widehat{\ell}\langle F(x_t) - F(x^*), x_t - x^*\rangle + \frac{1-q}{n}\sum_{i=1}^{n}\|F_i(w_t) - F_i(x^*)\|^2$$
$$= q\widehat{\ell}\langle F(x_t) - F(x^*), x_t - x^*\rangle + (1-q)\sigma_t^2,$$

here the second equality applies the definition of $w_{t+1}$, and the first inequality is implied by Assumption 3.3. The statement of Lemma 3.8 is obtained by comparing the above inequalities with the expressions of Assumption 2.2.  □

## F    Proofs of Main Convergence Results

### F.1    Proof of Theorem 3.1

Here the proof originates from that of ProxSkip (Mishchenko et al., 2022), while we extend it to the variational inequality setting, and combines it with the more general setting on the stochastic oracle.

*Proof.* Note that following Algorithm 1, we have with probability $p$:

$$\begin{cases} x_{t+1} = \mathbf{prox}_{\frac{\gamma}{p}R}\left(\widehat{x}_{t+1} - \frac{\gamma}{p}h_t\right) \\ h_{t+1} = h_t + \frac{p}{\gamma}\left(\mathbf{prox}_{\frac{\gamma}{p}R}\left(\widehat{x}_{t+1} - \frac{\gamma}{p}h_t\right) - \widehat{x}_{t+1}\right) \end{cases}$$

and with probability $1-p$:

$$\begin{cases} x_{t+1} = \widehat{x}_{t+1} \\ h_{t+1} = h_t, \end{cases}$$

and set

$$V_t \triangleq \|x_t - x^*\|^2 + \left(\frac{\gamma}{p}\right)^2\|h_t - F(x^*)\|^2 + M\gamma^2\sigma_t^2,$$

For simplicity, we denote $P(x_t) \triangleq \mathbf{prox}_{\frac{\gamma}{p}R}\left(\widehat{x}_{t+1} - \frac{\gamma}{p}h_t\right)$, so we have

$$\begin{aligned}
\mathbb{E}[V_{t+1}] &= p\left(\|P(x_t) - x^*\|^2 + \left(\frac{\gamma}{p}\right)^2\left\|h_t + \frac{p}{\gamma}(P(x_t) - \widehat{x}_{t+1}) - F(x^*)\right\|^2\right) \\
&\quad + (1-p)\left(\|\widehat{x}_{t+1} - x^*\|^2 + \left(\frac{\gamma}{p}\right)^2\|h_t - F(x^*)\|^2\right) + M\gamma^2\sigma_{t+1}^2 \\
&= p\left(\|P(x_t) - x^*\|^2 + \left\|P(x_t) - (\widehat{x}_{t+1} - \frac{\gamma}{p}h_t) - \frac{\gamma}{p}F(x^*)\right\|^2\right) \\
&\quad + (1-p)\left(\|\widehat{x}_{t+1} - x^*\|^2 + \left(\frac{\gamma}{p}\right)^2\|h_t - F(x^*)\|^2\right) + M\gamma^2\sigma_{t+1}^2
\end{aligned}$$

next note that $x^* = \mathbf{prox}_{\frac{\gamma}{p}R}\left(x^* - \frac{\gamma}{p}F(x^*)\right)$, we have

$$\begin{aligned}
&\left\|P(x_t) - (\widehat{x}_{t+1} - \frac{\gamma}{p}h_t) - \frac{\gamma}{p}F(x^*)\right\|^2 \\
&= \left\|P(x_t) - (\widehat{x}_{t+1} - \frac{\gamma}{p}h_t) - \left[\mathbf{prox}_{\frac{\gamma}{p}R}\left(x^* - \frac{\gamma}{p}F(x^*)\right) - (x^* - \frac{\gamma}{p}F(x^*))\right]\right\|^2
\end{aligned}$$

so by Lemma D.3, we have

$$\begin{aligned}
\mathbb{E}[V_{t+1}] &\overset{(10)}{\leq} p\left\|\widehat{x}_{t+1} - \frac{\gamma}{p}h_t - x^* + \frac{\gamma}{p}F(x^*)\right\|^2 + (1-p)\left(\|\widehat{x}_{t+1} - x^*\|^2 + \left(\frac{\gamma}{p}\right)^2\|h_t - F(x^*)\|^2\right) \\
&\quad + M\gamma^2\sigma_{t+1}^2 \\
&= \|\widehat{x}_{t+1} - x^*\|^2 + \left(\frac{\gamma}{p}\right)^2\|h_t - F(x^*)\|^2 - 2\frac{\gamma}{p}p\langle\widehat{x}_{t+1} - x^*, h_t - F(x^*)\rangle + M\gamma^2\sigma_{t+1}^2,
\end{aligned}$$

let

$$w_t \triangleq x_t - \gamma g(x_t), \quad w^* \triangleq x^* - \gamma F(x^*),$$

so we have

$$\begin{aligned}
&\|\widehat{x}_{t+1} - x^*\|^2 - 2\frac{\gamma}{p}p\langle\widehat{x}_{t+1} - x^*, h_t - F(x^*)\rangle \\
&= \|w_t - w^* + \gamma(h_t - F(x^*))\|^2 - 2\gamma\langle w_t - w^* + \gamma(h_t - F(x^*)), h_t - F(x^*)\rangle \\
&= \|w_t - w^*\|^2 - \gamma^2\|h_t - F(x^*)\|^2 \\
&= \|w_t - w^*\|^2 - p^2\left(\frac{\gamma}{p}\right)^2\|h_t - F(x^*)\|^2,
\end{aligned}$$

so we have

$$\mathbb{E}[V_{t+1}] \leq \|w_t - w^*\|^2 + (1 - p^2)\left(\frac{\gamma}{p}\right)^2 \|h_t - F(x^*)\|^2 + M\gamma^2\sigma_{t+1}^2. \tag{11}$$

Then by the standard analysis on GDA, we have

$$\begin{aligned}
\|w_t - w^*\|^2 &= \|x_t - x^* - \gamma(g(x_t) - F(x^*))\|^2 \\
&= \|x_t - x^*\|^2 - 2\gamma\langle g(x_t) - F(x^*), x_t - x^*\rangle + \gamma^2\|g(x_t) - F(x^*)\|^2,
\end{aligned}$$

take the expectation, by Assumption 2.2, we have

$$\begin{aligned}
\mathbb{E}\left[\|w_t - w^*\|^2\right] &= \|x_t - x^*\|^2 - 2\gamma\langle F(x_t) - F(x^*), x_t - x^*\rangle + \gamma^2\mathbb{E}\left[\|g(x_t) - F(x^*)\|^2\right] \\
&\leq \|x_t - x^*\|^2 - 2\gamma(1 - \gamma A)\langle F(x) - F(x^*), x - x^*\rangle + \gamma^2\left(B\sigma_t^2 + D_1\right),
\end{aligned}$$

substitute the above result into (11), we have

$$\begin{aligned}
\mathbb{E}[V_{t+1}] &\overset{(11)}{\leq} \mathbb{E}\left[\|w_t - w^*\|^2 + (1 - p^2)\left(\frac{\gamma}{p}\right)^2\|h_t - F(x^*)\|^2 + M\gamma^2\sigma_{t+1}^2\right] \\
&\leq \mathbb{E}\left[\|x_t - x^*\|^2 - 2\gamma(1 - \gamma A)\langle F(x) - F(x^*), x - x^*\rangle + \gamma^2\left(B\sigma_t^2 + D_1\right) \right. \\
&\qquad \left. + (1 - p^2)\left(\frac{\gamma}{p}\right)^2\|h_t - F(x^*)\|^2 + M\gamma^2\sigma_{t+1}^2\right] \\
&\leq \mathbb{E}\left[\|x_t - x^*\|^2 - 2\gamma(1 - \gamma(A + MC))\langle F(x) - F(x^*), x - x^*\rangle + M\gamma^2(1 - \rho)\sigma_t^2 \right. \\
&\qquad \left. + \gamma^2\left(B\sigma_t^2 + D_1 + MD_2\right) + (1 - p^2)\left(\frac{\gamma}{p}\right)^2\|h_t - F(x^*)\|^2\right] \\
&\leq \mathbb{E}\left[(1 - 2\gamma\mu(1 - \gamma(A + MC)))\|x_t - x^*\|^2 + (1 - p^2)\left(\frac{\gamma}{p}\right)^2\|h_t - F(x^*)\|^2 \right. \\
&\qquad \left. + M\gamma^2\left(1 - \rho + \frac{B}{M}\right)\sigma_t^2 + \gamma^2(D_1 + MD_2)\right],
\end{aligned}$$

here the third inequality comes from Assumption 2.2 on $\sigma_{t+1}^2$, and the fourth inequality comes from quasi-strong monotonicity in Assumption 2.1. Recall that $\gamma \leq \frac{1}{2(A+MC)}$, we have

$$\begin{aligned}
&\mathbb{E}[V_{t+1}] \\
&\leq \mathbb{E}\left[(1 - \gamma\mu)\|x_t - x^*\|^2 + (1 - p^2)\left(\frac{\gamma}{p}\right)^2\|h_t - F(x^*)\|^2 + \left(1 - \rho + \frac{B}{M}\right)M\gamma^2\sigma_t^2 + \gamma^2(D_1 + MD_2)\right] \\
&\leq \mathbb{E}\left[\left(1 - \min\left\{\gamma\mu, p^2, \rho - \frac{B}{M}\right\}\right)V_t\right] + \gamma^2(D_1 + MD_2),
\end{aligned}$$

by taking the full expectation and telescoping, we have

$$\begin{aligned}
\mathbb{E}[V_T] &\leq \left(1 - \min\left\{\gamma\mu, p^2, \rho - \frac{B}{M}\right\}\right)\mathbb{E}[V_{T-1}] + \gamma^2(D_1 + MD_2) \\
&= \left(1 - \min\left\{\gamma\mu, p^2, \rho - \frac{B}{M}\right\}\right)^T V_0 + \gamma^2(D_1 + MD_2)\sum_{i=0}^{T-1}\left(1 - \min\left\{\gamma\mu, p^2, \rho - \frac{B}{M}\right\}\right)^i \\
&\leq \left(1 - \min\left\{\gamma\mu, p^2, \rho - \frac{B}{M}\right\}\right)^T V_0 + \frac{\gamma^2(D_1 + MD_2)}{\min\left\{\gamma\mu, p^2, \rho - \frac{B}{M}\right\}},
\end{aligned}$$

the last inequality comes from the computation of a geometric series. which concludes the proof. Note that here we implicitly require that $\gamma \leq \frac{1}{\mu}$ and $M > \frac{B}{\rho}$. $\qquad\square$

### F.2 PROOF OF COROLLARY 3.2

*Proof.* With the above setting, we know that

$$\min\left\{\gamma\mu, p^2, \rho - \frac{B}{M}\right\} = \gamma\mu,$$

and

$$\mathbb{E}[V_T] \leq (1 - \gamma\mu)^T V_0 + \frac{\gamma\left(D_1 + \frac{2B}{\rho}D_2\right)}{\mu},$$

so it is easy to see that by setting

$$T \geq \frac{1}{\gamma\mu}\ln\left(\frac{2V_0}{\epsilon}\right), \quad \gamma \leq \frac{\mu\epsilon}{2\left(D_1 + \frac{2B}{\rho}D_2\right)},$$

we have

$$\mathbb{E}[V_T] \leq \epsilon,$$

which induces the iteration complexity to be

$$T \geq \max\left\{1, \frac{2(A + 2BC/\rho)}{\mu}, \frac{2}{\rho}, \frac{2\left(D_1 + \frac{2B}{\rho}D_2\right)}{\mu^2\epsilon}\right\}\ln\left(\frac{2V_0}{\epsilon}\right),$$

and the corresponding number of calls to the proximal oracle is

$$pT \geq \sqrt{\max\left\{1, \frac{2(A + 2BC/\rho)}{\mu}, \frac{2}{\rho}, \frac{2\left(D_1 + \frac{2B}{\rho}D_2\right)}{\mu^2\epsilon}\right\}\ln\left(\frac{2V_0}{\epsilon}\right)},$$

which concludes the proof. $\qquad\square$

### F.3 PROPERTIES OF OPERATORS IN VIPs

To make sure that the consensus form Problem (2) fits with Assumption 2.1, and the corresponding operator estimators satisfies Assumption 2.2, we provide the following results.

**Proposition F.1.** If Problem (1) attains an unique solution $z^* \in \mathbb{R}^{d'}$, then Problem (2) attains an unique solution $x^* \triangleq (z^*, z^*, \cdots, z^*) \in \mathbb{R}^d$, and vice versa.

*Proof.* Note that each $f_i$ is $\mu$-quasi-strongly monotone and $z^*$ is the unique solution to Problem (1). So for the operator $F$ in the reformulation (2), first we check the point $x^* = (z^*, z^*, \cdots, z^*)$, note that for any $x = (x_1, x_2, \cdots, x_n) \in \mathbb{R}^d$, we have

$$
\begin{aligned}
\langle F(x^*), x - x^*\rangle + R(x) - R(x^*) &= \left\langle \sum_{i=1}^n F_i(x^*), x - x^*\right\rangle + R(x) \\
&= \sum_{i=1}^n \langle F_i(x^*), x - x^*\rangle + R(x) \\
&= \sum_{i=1}^n \langle f_i(z^*), x_i - z^*\rangle + R(x),
\end{aligned}
$$

here the first equation incurs the definition of $R$, the third equation is due to the definition that $F_i$. Then for any $x \in \mathbb{R}^d$, if $\exists\, x_i \neq x_j$ for some $i, j \in [n]$, we have $R(x) = +\infty$, so the RHS above is always positive. Then if $x_1 = x_2 = \cdots = x_n = x' \in \mathbb{R}^{d'}$, we have

$$\sum_{i=1}^n \langle f_i(z^*), x_i - z^*\rangle + R(x) = \sum_{i=1}^n \langle f_i(z^*), x' - z^*\rangle = \left\langle \sum_{i=1}^n f_i(z^*), x' - z^*\right\rangle \geq 0,$$

where the last inequality comes from the fact that $z^*$ is the solution to Problem (1). So $x^*$ is the solution to Problem (2). It is easy to show its uniqueness by contradiction and the uniqueness of $z^*$, which we do not detail here.

On the opposite side, first it is easy to see that the solution to (2) must come with the form $x^* = (z^*, z^*, \cdots, z^*)$, then we have for any $x = (x_1, x_2, \cdots, x_n) \in \mathbb{R}^d$

$$\langle F(x^*), x - x^* \rangle + R(x) - R(x^*) = \sum_{i=1}^n \langle f_i(z^*), x_i - z^* \rangle + R(x),$$

then we select $x = (x', x', \cdots, x')$ for any $x' \in \mathbb{R}^{d'}$, so we have

$$\langle F(x^*), x - x^* \rangle + R(x) - R(x^*) = \sum_{i=1}^n \langle f_i(z^*), x' - z^* \rangle = \langle F(z^*), x' - z^* \rangle \geq 0,$$

which corresponds to the solution of (1), and concludes the proof. $\qquad \square$

**Proposition F.2.** With Assumption 4.1, the Problem (2) attains an unique solution $x^* \triangleq (z^*, z^*, \cdots, z^*) \in \mathbb{R}^d$, the operator $F$ is $\mu$-quasi-strongly monotone, and the operator $g(x) \triangleq (g_1(x_1), g_2(x_2), \cdots, g_n(x_n))$ is an unbiased estimator of $F$, and it satisfies $L_g$-expected cocoercivity defined in Assumption 2.2.

*Proof.* The uniqueness result comes from the above proposition. Then note that

$$
\begin{aligned}
\langle F(x) - F(x^*), x - x^* \rangle &= \left\langle \sum_{i=1}^n (F_i(x) - F_i(x^*)), x - x^* \right\rangle \\
&= \sum_{i=1}^n \langle F_i(x) - F_i(x^*), x - x^* \rangle \\
&= \sum_{i=1}^n \langle f_i(x_i) - f_i(z^*), x_i - z^* \rangle \\
&\geq \sum_{i=1}^n \mu \|x_i - z^*\|^2 \\
&= \mu \|x - x^*\|^2,
\end{aligned}
$$

which verifies the second statement. For the last statement, following the definition of the estimator $g$,

$$
\mathbb{E}g(x^*) = \mathbb{E} \begin{pmatrix} g_1(z^*) \\ g_2(z^*) \\ \vdots \\ g_n(z^*) \end{pmatrix} = \begin{pmatrix} f_1(z^*) \\ f_2(z^*) \\ \vdots \\ f_n(z^*) \end{pmatrix} = \sum_{i=1}^n F_i(x^*) = F(x^*),
$$

which implies that it is an unbiased estimator of $F$. Then note that for any $x \in \mathbb{R}^d$,

$$
\begin{aligned}
\mathbb{E}\|g(x) - g(x^*)\|^2 &= \mathbb{E} \sum_{i=1}^n \|(g_i(x_i) - g_i(z^*))\|^2 \\
&= \sum_{i=1}^n \mathbb{E}\|g_i(x_i) - g_i(z^*)\|^2 \\
&\leq \sum_{i=1}^n L_g \langle f_i(x_i) - f_i(z^*), x_i - z^* \rangle \\
&= L_g \sum_{i=1}^n \langle F_i(x) - F_i(x^*), x - x^* \rangle \\
&= L_g \langle F(x) - F(x^*), x - x^* \rangle,
\end{aligned}
$$

here the inequality comes from the expected cocoercivity of each $g_i$ in Assumption 4.1, which concludes the proof. $\qquad \square$

## F.4 PROPERTIES OF OPERATORS IN FINITE-SUM VIPs

**Proposition F.3.** With Assumption 4.3, the problem defined above attains an unique solution $x^* \triangleq (z^*, z^*, \cdots, z^*) \in \mathbb{R}^d$, the operator $F$ is $\frac{\mu}{n}$-quasi-strongly monotone, and the operator $g_t \triangleq (g_{1,t}, g_{2,t}, \cdots, g_{n,t})$ satisfies Assumption 2.2 with

$$A = \widehat{\ell}, \ B = 2, \ C = \frac{q\widehat{\ell}}{2}, \ \rho = q, \ D_1 = D_2 = 0, \ \sigma_t^2 = \sum_{i=1}^{n} \frac{1}{m_i} \sum_{j=1}^{m_i} \|F_{i,j}(z^*) - F_{i,j}(w_{i,t})\|^2.$$

*Proof.* Here the proof is similar to that of Lemma 3.8. The conclusions on the solution $x^*$ and the quasi-strong monotonicity follow the same argument in the proof of Proposition F.2. For the property of $g_t$, it is easy to find that the unbiasedness holds. Then note that

$$\mathbb{E}\|g_t - F(x^*)\|^2$$
$$= \sum_{i=1}^{n} \mathbb{E}\|g_{i,t} - F_i(x^*)\|^2$$
$$= \sum_{i=1}^{n} \mathbb{E}\|F_{i,j_t}(x_{i,t}) - F_{i,j_t}(w_{i,t}) + F_i(w_{i,j_t}) - F_i(z^*)\|^2$$
$$= \sum_{i=1}^{n} \left( \frac{1}{m_i} \sum_{j=1}^{m_i} \|F_{i,j}(x_{i,t}) - F_{i,j}(w_{i,t}) + F_i(w_{i,t}) - F_i(z^*)\|^2 \right)$$
$$\overset{(7)}{\leq} \sum_{i=1}^{n} \left( \frac{2}{m_i} \sum_{j=1}^{m_i} \|F_{i,j}(x_{i,t}) - F_{i,j}(z^*)\|^2 + \|F_{i,j}(z^*) - F_{i,j}(w_{i,t}) - (F_i(z^*) - F_i(w_{i,t}))\|^2 \right)$$
$$\leq 2 \sum_{i=1}^{n} \left( \frac{1}{m_i} \sum_{j=1}^{m_i} \|F_{i,j}(x_{i,t}) - F_{i,j}(z^*)\|^2 + \|F_{i,j}(z^*) - F_{i,j}(w_{i,t})\|^2 \right)$$
$$\leq 2\widehat{\ell} \sum_{i=1}^{n} \langle f_i(x_{i,t}) - f_i(z^*), x_{i,t} - z^* \rangle + 2 \sum_{i=1}^{n} \frac{1}{m_i} \sum_{j=1}^{m_i} \|F_{i,j}(z^*) - F_{i,j}(w_{i,t})\|^2$$
$$= 2\widehat{\ell} \sum_{i=1}^{n} \langle F_i(x_{i,t}) - F_i(z^*), x_t - x^* \rangle + 2 \sum_{i=1}^{n} \frac{1}{m_i} \sum_{j=1}^{m_i} \|F_{i,j}(z^*) - F_{i,j}(w_{i,t})\|^2$$
$$= 2\widehat{\ell} \langle F(x_t) - F(x^*), x_t - x^* \rangle + 2\sigma_t^2,$$

the second inequality comes from the fact that $\mathrm{Var}(X) \leq \mathbb{E}(X^2)$, and the third inequality is implied by Assumption 4.3. Then for the second term above, we have

$$\mathbb{E}[\sigma_{t+1}^2] = \sum_{i=1}^{n} \frac{1}{m_i} \sum_{j=1}^{m_i} \mathbb{E}\|F_{i,j}(w_{i,t+1}) - F_{i,j}(z^*)\|^2$$
$$= \sum_{i=1}^{n} \frac{1}{m_i} \sum_{j=1}^{m_i} \left( q\|F_{i,j}(x_{i,t}) - F_{i,j}(z^*)\|^2 + (1-q)\|F_{i,j}(w_{i,t}) - F_{i,j}(z^*)\|^2 \right)$$
$$\leq \sum_{i=1}^{n} q\widehat{\ell} \langle f_i(x_{i,t}) - f_i(z^*), x_{i,t} - z^* \rangle + (1-q) \sum_{i=1}^{n} \frac{1}{m_i} \sum_{j=1}^{m_i} \|F_{i,j}(w_{i,t}) - F_{i,j}(z^*)\|^2$$
$$= q\widehat{\ell} \langle F(x_t) - F(x^*), x_t - x^* \rangle + (1-q)\sigma_t^2,$$

here the second equality comes from the definition of $w_{i,t+1}$, and the inequality comes from Assumption 4.3. So we conclude the proof. $\square$

## F.5 FURTHER COMPARISON OF COMMUNICATION COMPLEXITIES

In Table 2, following the discussion ("Comparison with Literature") in Section 4, we compare our convergence results to closely related works on stochastic local training methods. The Table shows the

improvement in terms of communication and iteration complexities of our proposed ProxSkip-VIP-FL algorithms over methods like Local SGDA (Deng & Mahdavi, 2021), Local SEG (Beznosikov et al., 2020) and FedAvg-S (Hou et al., 2021).

Table 2: Comparison of federated learning algorithms for solving VIPs with strongly monotone and Lipschitz operator. Comparison is in terms of both iteration and communication complexities.

| Algorithm | Setting[1] | # Communication[2] | # Iteration |
|---|---|---|---|
| **Local SGDA** (Deng & Mahdavi, 2021) | SM, LS | $\mathcal{O}\left(\sqrt{\frac{\kappa^2\sigma_*^2}{\mu\epsilon}}\right)$ | $\mathcal{O}\left(\frac{\kappa^2\sigma_*^2}{\mu n\epsilon}\right)$ |
| **Local SEG** (Beznosikov et al., 2020) | SM, LS | $\mathcal{O}\left(\max\left(\kappa\ln\frac{1}{\epsilon},\frac{p\Delta^2}{\mu^2 n\epsilon},\frac{\kappa\xi}{\mu\sqrt{\epsilon}},\frac{\sqrt{p}\kappa\Delta}{\mu\sqrt{\epsilon}}\right)\right)$ | $\mathcal{O}\left(\max\left(\frac{\kappa}{p}\ln\frac{1}{\epsilon},\frac{\Delta^2}{\mu^2 n\epsilon},\frac{\kappa\xi}{p\mu\sqrt{\epsilon}},\frac{\kappa\Delta}{\mu\sqrt{p\epsilon}}\right)\right)$ |
| **FedAvg-S** (Hou et al., 2021) | SM, LS | $\tilde{\mathcal{O}}\left(\frac{p\Delta^2}{n\mu^2\epsilon}+\frac{\sqrt{p}\kappa\Delta}{\mu\sqrt{\epsilon}}+\frac{\kappa\xi}{\mu\sqrt{\epsilon}}\right)$ | $\tilde{\mathcal{O}}\left(\frac{\Delta^2}{n\mu^2\epsilon}+\frac{\kappa\Delta}{\mu\sqrt{p\epsilon}}+\frac{\kappa\xi}{p\mu\sqrt{\epsilon}}\right)$ |
| **ProxSkip-VIP-FL** (This work) | SM, LS[3] | $\mathcal{O}\left(\sqrt{\max\left\{\kappa^2,\frac{\sigma_*^2}{\mu^2\epsilon}\right\}}\ln\frac{1}{\epsilon}\right)$ | $\mathcal{O}\left(\max\left\{\kappa^2,\frac{\sigma_*^2}{\mu^2\epsilon}\right\}\ln\frac{1}{\epsilon}\right)$ |
| **ProxSkip-L-SVRGDA-FL** (This work)[4] | SM, LS | $\mathcal{O}\left(\kappa\ln\frac{1}{\epsilon}\right)$ | $\mathcal{O}\left(\kappa^2\ln\frac{1}{\epsilon}\right)$ |

[1] SM: strongly monotone, LS: (Lipschitz) smooth. $\kappa \triangleq L/\mu$, $L$ and $\mu$ are the modulus of SM and LS. $\sigma_*^2 < +\infty$ is an upper bound of the variance of the stochastic operator at $x^*$. $\Delta$ is an (uniform) upper bound of the variance of the stochastic operator. $\xi^2$ represents the bounded heterogeneity, i.e., $g_i(x;\xi_i)$ is an unbiased estimator of $f_i(x)$ for any $i \in \{1,\ldots,n\}$, and $\xi_i^2(x) \triangleq \sup_{x\in\mathbb{R}^d}\|f_i(x)-F(x)\|^2 \le \xi^2 \le +\infty$.

[2] $p$ is the probability of synchronization, we can take $p = \mathcal{O}(\sqrt{\epsilon})$, which recovers $\mathcal{O}(1/\sqrt{\epsilon})$ communication complexity dependence on $\epsilon$ in our result. $\tilde{\mathcal{O}}(\cdot)$ hides the logarithmic terms.

[3] Our algorithm works for quasi-strongly monotone and star-cocoercive operators, which is more general than the SM and LS setting, note that an $L$-LS and $\mu$-SM operator can be shown to be $(\kappa L)$-star-cocoercive (Loizou et al., 2021).

[4] When we further consider the finite-sum form problems, we can turn to this algorithm.

## G    DETAILS ON NUMERICAL EXPERIMENTS

In experiments, we examine the performance of ProxSkip-VIP-FL and ProxSkip-L-SVRGDA-FL. We compare ProxSkip-VIP-FL and ProxSkip-L-SVRGDA-FL algorithm with Local SGDA (Deng & Mahdavi, 2021) and Local SEG (Beznosikov et al., 2020), and the parameters are chosen according to corresponding theoretical convergence guarantees. Given any function $f(x_1, x_2)$, the $\ell$ co-coercivity parameter of the operator

$$\begin{pmatrix} \nabla_{x_1} f(x_1, x_2) \\ -\nabla_{x_2} f(x_1, x_2) \end{pmatrix}$$

is given by $\frac{1}{\ell} = \min_{\lambda \in \mathrm{Sp}(J)} \mathcal{R}\left(\frac{1}{\lambda}\right)$ (Loizou et al., 2021). Here, Sp denotes spectrum of the Jacobian matrix

$$J = \begin{pmatrix} \nabla^2_{x_1, x_1} f & \nabla^2_{x_1, x_2} f \\ -\nabla^2_{x_1, x_2} f & -\nabla^2_{x_2, x_2} f \end{pmatrix}.$$

In our experiment, for the min-max problem of the form

$$\min_{x_1} \max_{x_2} \frac{1}{n} \sum_{i=1}^{n} \frac{1}{m_i} \sum_{j=1}^{m_i} f_{ij}(x_1, x_2), \tag{12}$$

we use stepsizes according to the following Table.

| Algorithm | Stepsize $\gamma$ | Value of $p$ |
|:---:|:---:|:---:|
| **ProxSkip-VIP-FL** (deterministic) | $\gamma = \frac{1}{2 \max_{i \in [n]} \ell_i}$ | $p = \sqrt{\gamma \mu}$ |
| **ProxSkip-VIP-FL** (stochastic) | $\gamma = \frac{1}{2 \max_{i,j} \ell_{ij}}$ | $p = \sqrt{\gamma \mu}$ |
| **ProxSkip-L-SVRGDA-FL** (finite-sum) | $\gamma = \frac{1}{6 \max_{i,j} \ell_{ij}}$ | $p = \sqrt{\gamma \mu}$ |

Table 3: Parameter settings for each algorithm. Here $\ell_i$ is the co-coercivity parameter corresponding to $\frac{1}{m_i} \sum_{j=1}^{m_i} f_{ij}(x_1, x_2)$ and $\ell_{ij}$ is the co-coercivity parameter corresponding to $f_{ij}$.

The parameters in Table 3 are selected based on our theoretical convergence guarantees, presented in the main paper. In particular, for ProxSkip-VIP-FL, we use the stepsizes suggested in Theorem 4.2, which follows the setting of Corollary 3.5. Note that the full-batch estimator (deterministic setting) of (12) satisfies Assumption 4.1 when $L_g = \max_{i \in [n]} \ell_i$. Similarly, the stochastic estimators of (12) satisfies Assumption 4.1 with $L_g = \max_{i,j} \ell_{ij}$. For the variance reduced method, ProxSkip-L-SVRGDA-FL, we use the stepsizes as suggested in Theorem 4.4 (which follows the setting in Corollary 3.9) with $\hat{\ell} = \max_{i,j} \ell_{ij}$ for (12). For all methods, the probability of making the proximal update (communication in the FL regime) equals $p = \sqrt{\gamma \mu}$. For the ProxSkip-L-SVRGDA-FL, following Corollary 3.9 we set $q = 2\gamma \mu$.

### G.1    DETAILS ON ROBUST LEAST SQUARES

The objective function of Robust Least Square is given by

$$G(\beta, y) = \|\mathbf{A}\beta - y\|^2 - \lambda\|y - y_0\|^2,$$

for $\lambda > 0$. Note that

$$\nabla^2_\beta G(\beta, y) = 2\mathbf{A}^\top \mathbf{A}, \quad \nabla^2_y G(\beta, y) = 2(\lambda - 1)\mathbf{I}.$$

Therefore, for $\lambda > 1$, the objective function $G$ is strongly monotone with parameter $\min\{2\lambda_{\min}(\mathbf{A}^\top \mathbf{A}), 2(\lambda - 1)\}$. Moreover, $G$ can be written as a finite sum problem, similar to (5), by decomposing the rows of matrix $\mathbf{A}$ i.e.

$$G(\beta, y) = \sum_{i=1}^{r} (A_i^\top \beta - y_i)^2 - \lambda(y_i - y_{0i})^2.$$

Here $\mathbf{A}_i^\top$ denotes the $i$th row of the matrix $\mathbf{A}$. Now we divide the $r$ rows among $n$ nodes where each node will have $m = r/n$ rows. Then we use the canonical basis vectors $e_i$ (vector with $i$-th entry 1 and 0 for other coordinates) to rewrite the above problem as follow

$$
\begin{aligned}
G(\beta, y) &= \sum_{i=1}^{n} \sum_{j=1}^{m} \left( A_{(i-1)m+j}^\top \beta - y_{(i-1)m+j} \right)^2 - \lambda \left( y_{(i-1)m+j} - y_{0((i-1)m+j)} \right)^2 \\
&= \sum_{i=1}^{n} \sum_{j=1}^{m} \beta^\top A_{(i-1)m+j} A_{(i-1)m+j}^\top \beta - 2 y_{(i-1)m+j} A_{(i-1)m+j}^\top \beta + y_{(i-1)m+j}^2 \\
&\quad - \lambda y_{(i-1)m+j}^2 - \lambda y_{0((i-1)m+j)}^2 + 2\lambda y_{0((i-1)m+j)} y_{((i-1)m+j)} \\
&= \sum_{i=1}^{n} \sum_{j=1}^{m} \beta^\top \left( A_{(i-1)m+j} A_{(i-1)m+j}^\top \right) \beta - y^\top \left( 2 e_{(i-1)m+j} A_{(i-1)m+j}^\top \right) \beta \\
&\quad - y^\top \left( (\lambda - 1) e_{(i-1)m+j} e_{(i-1)m+j}^\top \right) y + \left( 2\lambda y_{0((i-1)m+j)} e_{((i-1)m+j)}^\top \right) y \\
&\quad - \lambda y_{0((i-1)m+j)}^2 \\
&= \frac{nm}{n} \sum_{i=1}^{n} \frac{1}{m} \sum_{j=1}^{m} \beta^\top \left( A_{(i-1)m+j} A_{(i-1)m+j}^\top \right) \beta - y^\top \left( 2 e_{(i-1)m+j} A_{(i-1)m+j}^\top \right) \beta \\
&\quad - y^\top \left( (\lambda - 1) e_{(i-1)m+j} e_{(i-1)m+j}^\top \right) y + \left( 2\lambda y_{0((i-1)m+j)} e_{((i-1)m+j)}^\top \right) y \\
&\quad - \lambda y_{0((i-1)m+j)}^2 .
\end{aligned}
$$

Therefore $G$ is equivalent to (5) with $n$ nodes, $m_i = m = r/n, x_1 = \beta, x_2 = y$ and

$$
\begin{aligned}
f_{ij}(x_1, x_2) &= x_1^\top \left( A_{(i-1)m+j} A_{(i-1)m+j}^\top \right) x_1 - x_2^\top \left( 2 e_{(i-1)m+j} A_{(i-1)m+j}^\top \right) x_1 \\
&\quad - x_2^\top \left( (\lambda - 1) e_{(i-1)m+j} e_{(i-1)m+j}^\top \right) x_2 + \left( 2\lambda y_{0((i-1)m+j)} e_{((i-1)m+j)}^\top \right) x_2 \\
&\quad - \lambda y_{0((i-1)m+j)}^2 .
\end{aligned}
$$

In Figure 2, we run our experiment on the "California Housing" dataset from scikit-learn package (Pedregosa et al., 2011). This data consists of 8 attributes of 200 houses in the California region where the target variable $y_0$ is the price of the house. To implement the algorithms, we divide the data matrix $\mathbf{A}$ among 20 nodes, each node having an equal number of rows of $\mathbf{A}$. Similar to the last example, here we also choose our ProxSkip-VIP-FL algorithm, Local SGDA, and Local SEG for comparison in the experiment, also we use $\lambda = 50$, and the theoretical stepsize choice is similar to the previous experiment.

In Figure 3, we reevaluate the performance of ProxSkip on the Robust Least Square problem with synthetic data. For generating the synthetic dataset, we set $r = 200$ and $s = 20$. Then we sample $\mathbf{A} \sim \mathcal{N}(0, 1), \beta_0 \sim \mathcal{N}(0, 0.1), \epsilon \sim \mathcal{N}(0, 0.01)$ and set $y_0 = \mathbf{A}\beta_0 + \epsilon$. In both deterministic (Figure 3a) and stochastic (Figure 3b) setting, ProxSkip outperforms Local GDA and Local EG.

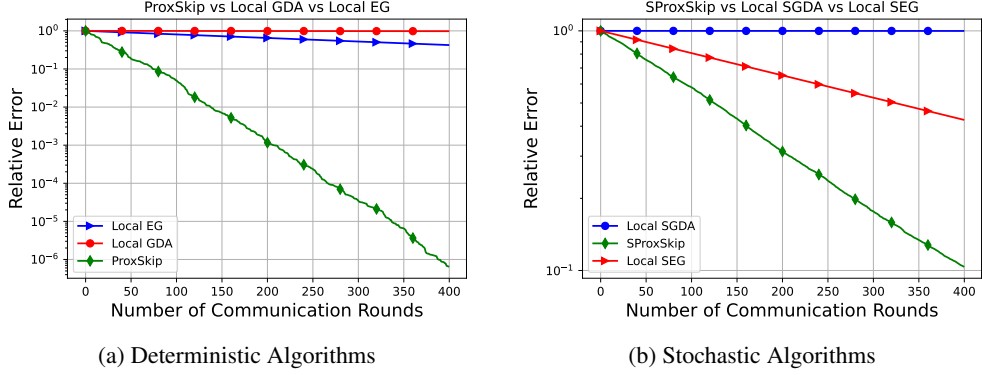

(a) Deterministic Algorithms        (b) Stochastic Algorithms

Figure 3: Comparison of algorithms on the Robust Least Square (6) using synthetic dataset.

## H ADDITIONAL EXPERIMENTS

Following the experiments presented in the main paper, we further evaluate the performance of the proposed methods in different settings (problems and stepsize selections).

### H.1 FINE-TUNED STEPSIZE

In Figure 4, we compare the performance of ProxSkip against that of Local GDA and Local EG on the strongly monotone quadratic game (5) with heterogeneous data using tuned stepsizes. For tuning the stepsizes, we did a grid search on the set of $\frac{1}{rL}$ where $r \in \{1, 2, 4, 8, 16, 64, 128, 256, 512, 1024, 2048\}$ and $L$ is the Lipschitz constant of $F$. ProxSkip outperforms the other two methods in the deterministic setting while it has a comparable performance in the stochastic setting.

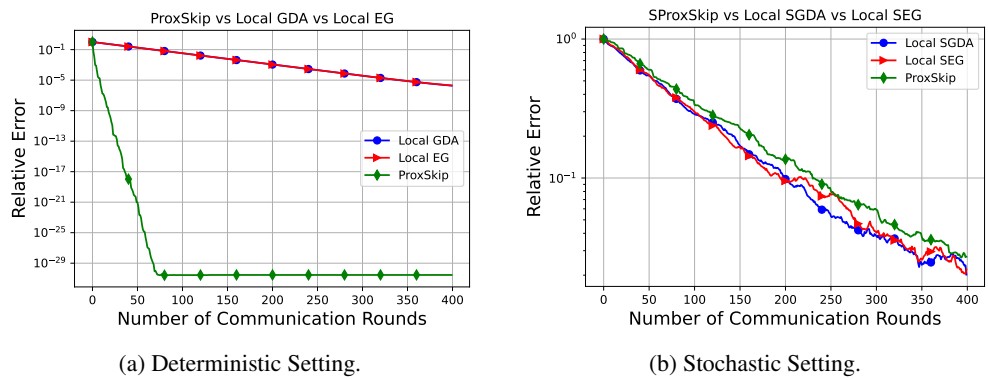

(a) Deterministic Setting.    (b) Stochastic Setting.

Figure 4: *Comparison of ProxSkip-VIP-FL vs Local SGDA vs Local SEG on Heterogeneous Data with tuned stepsizes.*

### H.2 PROXSKIP-VIP-FL VS. PROXSKIP-L-SVRGDA-FL

In Figure 5, we compare the stochastic version of ProxSkip-VIP-FL with ProxSkip-L-SVRGDA-FL. In Figure 5a, we implement the methods with tuned stepsizes while in Figure 5b we use the theoretical stepsizes. For the theoretical stepsizes of ProxSkip-L-SVRGDA-FL, we use the stepsizes from Corollary 3.9. We observe that ProxSkip-L-SVRGDA-FL performs better than ProxSkip-VIP-FL with both tuned and theoretical stepsizes.

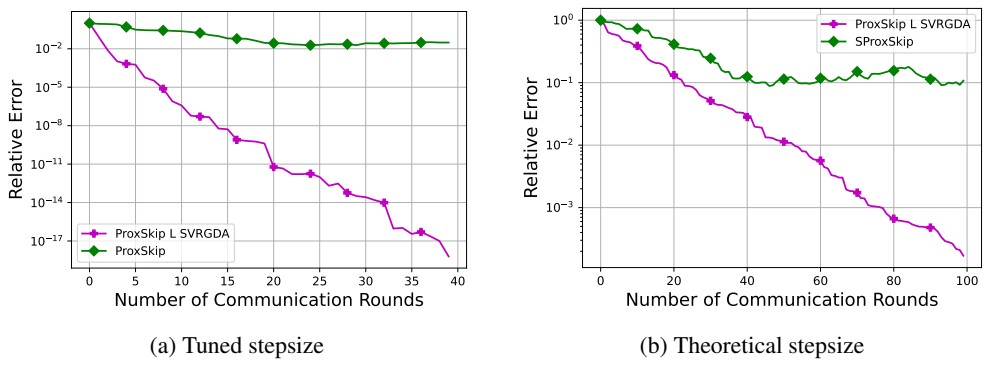

(a) Tuned stepsize    (b) Theoretical stepsize

Figure 5: *Comparison of ProxSkip-VIP-FL and ProxSkip-L-SVRGDA-FL using the tuned and theoretical stepsizes.*

### H.3 Low vs High Heterogeneity

We conduct a numerical experiment on a toy example to verify the efficiency of our proposed algorithm. Following the setting in (Tarzanagh et al., 2022), we consider the minimax objective function

$$\min_{x_1 \in \mathbb{R}^d} \max_{x_2 \in \mathbb{R}^d} \frac{1}{n} \sum_{i=1}^{n} f_i(x_1, x_2)$$

where $f_i$ are given by

$$f_i(x_1, x_2) = -\left[\frac{1}{2}\|x_2\|^2 - b_i^\top x_2 + x_2^\top A_i x_1\right] + \frac{\lambda}{2}\|x_1\|^2$$

Here we set the number of clients $n = 100$, $d_1 = d_2 = 20$ and $\lambda = 0.1$. We generate $b_i \sim \mathcal{N}(0, s_i^2 I_{d_2})$ and $A_i = t_i I_{d_1 \times d_2}$. For Figure 6a, we set $s_i = 10$ and $t_i = 1$ for all $i$ while in Figure 6b, we generate $s_i \sim \text{Unif}(0, 20)$ and $t_i \sim \text{Unif}(0, 1)$.

We implement Local GDA, Local EG, and ProxSkip-VIP-FL with tuned stepsizes (we use grid search to tune stepsizes Appendix G). In Figure 6c, we observe that Local EG performs better than ProxSkip-VIP-FL in homogeneous data. However, in Figure 6d, the performance of Local GDA (Deng & Mahdavi, 2021) and Local EG deteriorates for heterogeneous data, and ProxSkip-VIP-FL outperforms both of them in this case. To get stochastic estimates, we add Gaussian noise (Beznosikov et al., 2022a) (details in Appendix G). We repeat this experiment in the stochastic settings in Figure 6c and 6d. We observe that ProxSkip-VIP-FL has a comparable performance with Local SGDA (Deng & Mahdavi, 2021) and Local SEG in homogeneous data. However, ProxSkip-VIP-FL is faster on heterogeneous data.

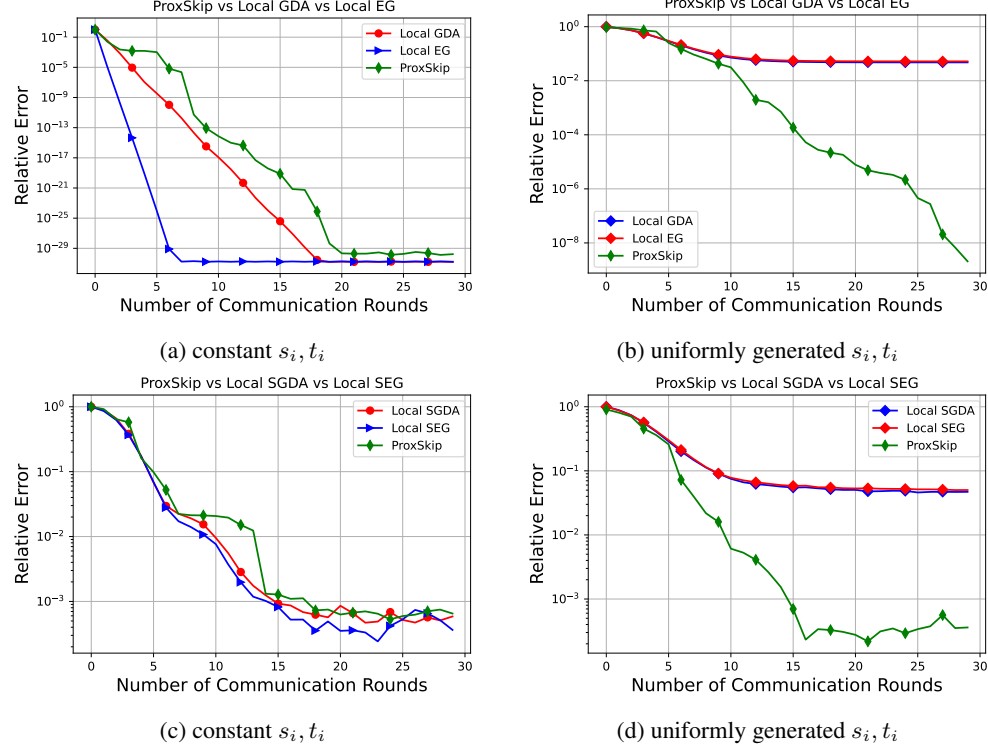

(a) constant $s_i, t_i$

(b) uniformly generated $s_i, t_i$

(c) constant $s_i, t_i$

(d) uniformly generated $s_i, t_i$

Figure 6: *Comparison of ProxSkip-VIP-FL vs Local GDA vs Local EG on Homogeneous vs Heterogeneous Data. In (a) and (b), we run the deterministic algorithms, while in (c) and (d), we run the stochastic versions. For (a) and (c), we set $s_i = 10, t_i = 1$ for all $i \in [n]$ and for (b) and (d), we generate $s_i, t_i$ uniformly from $s_i \sim \text{Unif}(0, 20), t_i \sim \text{Unif}(0, 1)$.*

### H.4 PERFORMANCE ON DATA WITH VARYING HETEROGENEITY

In this experiment, we consider the operator $F$ given by

$$F(x) := \frac{1}{2}F_1(x) + \frac{1}{2}F_2(x)$$

where

$$F_1(x) := M(x - x_1^*), \quad F_2(x) := M(x - x_2^*)$$

with $M \in \mathbb{R}^{2 \times 2}$ and $x_1^*, x_2^* \in \mathbb{R}^2$. For this experiment we choose

$$M := I_2, \quad x_1^* = (\delta, 0), \quad x_2^* = (0, \delta).$$

Note that, in this case, $x^* = \frac{1}{2}(x_1^* + x_2^*)$. Then the quantity $\max_{i \in [2]} \|F_i(x^*) - F(x^*)\|^2$, which quantifies the amount of heterogeneity in the model, is equal to $\frac{\delta^2}{2}$. Therefore, increasing the value of $\delta$ increases the amount of heterogeneity in the data across the clients.

We compare the performances of ProxSkip-VIP-FL, Local GDA, and Local EG when $\delta = 0$ and $\delta = 10^6$ in Figure 7. In either case, ProxSkip-VIP-FL outperforms the other two methods with the theoretical stepsizes.

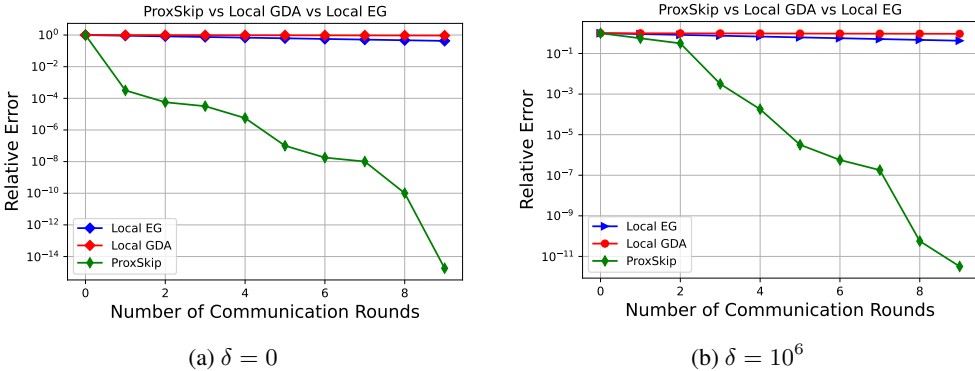

(a) $\delta = 0$            (b) $\delta = 10^6$

Figure 7: *Comparison of ProxSkip-VIP-FL, Local GDA and Local EG with theoretical stepsizes for $\delta = 0$ (left) and $\delta = 10^6$ (right).*

### H.5 EXTRA EXPERIMENT: POLICEMEN BURGLAR PROBLEM

In this experiment, we compare the perforamnce of ProxSkip-GDA-FL, Local GDA and Local EG on a Policemen Burglar Problem (Nemirovski, 2013) (a particular example of matrix game) of the form:

$$\min_{x_1 \in \Delta} \max_{x_2 \in \Delta} f(x_1, x_2) = \frac{1}{n} \sum_{i=1}^{n} x_1^\top A_i x_2$$

where $\Delta = \left\{ x \in \mathbb{R}^d \mid \mathbf{1}^\top x = 1, x \geq 0 \right\}$ is a $(d-1)$ dimensional standard simplex. We generate the $(r, s)$-th element of the matrix $A_i$ as follow

$$A_i(r, s) = w_r \left( 1 - \exp \left\{ -0.8|r - s| \right\} \right) \qquad \forall i \in [n]$$

where $w_r = |w_r'|$ with $w_r' \sim \mathcal{N}(0, 1)$. This matrix game is a constrained monotone problem and we use duality gap to measure the performance of the algorithms. Note that the duality gap for the problem $\min_{x \in \Delta} \max_{y \in \Delta} f(x, y)$ at $(\hat{x}, \hat{y})$ is defined as $\max_{y \in \Delta} f(\hat{x}, y) - \min_{x \in \Delta} f(x, \hat{y})$.

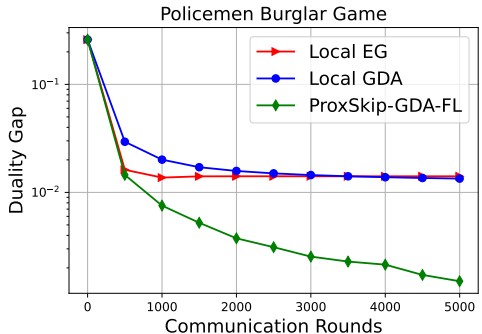

Figure 8: Comparison of ProxSkip-GDA-FL, Local EG and Local GDA on Policemen Burglar Problem after 5000 communication rounds.

In Figure 8, we plot the duality gap (on the $y$-axis) with respect to the moving average of the iterates, i.e. $\frac{1}{K+1} \sum_{k=0}^{K} x_k$ (here $x_k$ is the output after $k$ many communication rounds). As we can observe in Figure 8, our proposed algorithm ProxSkip-GDA-FL clearly outperforms Local EG and Local GDA in this experiment.

