# OpenReview forum: "Communication-Efficient Gradient Descent-Accent Methods for Distributed Variational Inequalities: Unified Analysis and Local Updates"
_ICLR.cc/2024/Conference — ICLR 2024 poster_

### Official Review · Reviewer_HQew · 2023-10-25

**Soundness:** 3 good
**Presentation:** 3 good
**Contribution:** 2 fair
**Rating:** 6
**Confidence:** 3

**Summary:**

This paper studies federated minimax optimization problems.

The authors proposed Proxskip-GDA and Proskip-L-SVRGDA which generalize the recent advance of Proxskip (Mishchenko et.al 2022) on the minimax (variational inequalities) problems and establish the new state-of-art in terms of communication complexity in both deterministic and stochastic setting.

However, the analysis and design of the proposed methods are very similar to the ones for minization problems, including the deterministic setting (Mishchenko et.al 2022) and the stochastic setting (Malinovsky et.al 2022). The author need to clarify the differences between their methods and the previous Proxskip framework (especially the hardness when generalize these methods from minimization to minimax).

**Strengths:**

The convergence rates in this paper are significant. They cover the settings of both stochastic and deterministic and establish the new SOTA.

**Weaknesses:**

My main concern about this paper is its novelty. The generalization of proxskip framwork (Mishchenko et.al 2022) into minimax optimization seems direct. The variance reduction variant is interesting, but the technique is also similar to the work for minimization problems (Malinovsky et.al 2022).

For the experimental part, the author should also compare their methods with Sun et.al 2022 which firstly establish the linear rate under similar settings.

**Questions:**

Please refer to the weakness part.

---

> ### Author Response · Authors · 2023-11-22
> **Response to Reviewer HQew**
>
> Thank you very much for your time and insightful comments.
>
> We thank the Reviewer HQew for appreciating the convergence guarantees provided in our work for different settings (deterministic and stochastic). Moreover, we are glad that HQew finds the variance reduction variant of our algorithm interesting.
> We address your concerns as follows.
>
> ---
> **Question 1**: My main concern about this paper is its novelty. The generalization of proxskip framwork (Mishchenko et.al 2022) into minimax optimization seems direct. The variance reduction variant is interesting, but the technique is also similar to the work for minimization problems.
>
> **Response 1**: Thank you for the comments. We respectfully express a differing viewpoint regarding the originality of our results. While we agree that our work is inspired by Mishchenko et al., 2022 and Malinovsky et.al 2022, we have significantly expanded upon their analysis in various key aspects. This divergence primarily stems from our critical Assumption 2.2, which plays a pivotal role in defining the behavior of operator estimation. This assumption necessitates the incorporation of $M\gamma^2 \sigma_t^2$ in our Lyapunov function, leading to a distinctly different analytical approach in our work. At the same time, we need to incorporate convergence analysis tricks appearing in the gradient descent-ascent (GDA) literature to derive the final convergence result.
>
> An important point to we want to highlight is that the previous works in simple optimization problems involve function suboptimality (a concept that cannot be useful in VI problems) in their proof techniques. In our proofs, there are no function values. Thus, the difference between the two pieces of literature does not solely lie in different settings (minimization vs VIPs) but begins at a deeper conceptual level.
>
> Finally, let us highlight that from a significance perspective, our work provides a unified algorithm framework under a very general estimator setting in the VIP regime; also, besides improved communication complexity, we are the first to provide an algorithm that avoids the bounded heterogeneity assumption. **For the above points, we respectfully stand by our claim of novelty and believe this is not a reason for suggesting rejection.**
>
> ---
> **Question 2**: For the experimental part, the author should also compare their methods with Sun et.al 2022 which firstly establish the linear rate under similar settings.
>
> **Response 2**: Thank you very much for the pointer. As suggested, we have incorporated the FedGDA-GT algorithm from Sun et al. (2022) into the comparison and attached the new result [(HERE)](https://ibb.co/PMwcfwT).
>
> Let us also highlight some critical differences in theoretical results between our work and Sun et al. (2022):
> - In terms of convergence guarantees, the algorithm the FedGDA-GT algorithm from Sun et al. (2022) converges only in the full batch setting (deterministic), while our approach and algorithms handle the more challenging stochastic scenarios as well. In addition, our algorithm avoids the compact constraints in the function (a requirement in Sun et al. (2022)).
> - As we can see in the attached plot in the deterministic strongly-monotone quadratic game (Eq.6), our algorithm still outperforms all other candidates, including FedGDA-GT, which further rationalizes the empirical benefits of our proposed ProxSkip-GDA-FL algorithm. Finally, let us note that ProxSkip-GDA-FL enjoys acceleration in terms of communication complexity for solving strongly monotone problems (see our main theorems), which is not the case in Sun et al. (2022). This can explain the reason why ProxSkip-GDA-FL performs better than FedGDA-GT in the experiments.
> - In addition, as we mentioned in Appendix A of our paper, the gradient tracking's (the technique used by Sun et al. (2022))  provable communication complexity scales linearly in the
> condition number even when combined with local steps. This does not happen with our methods. In the camera-ready version of our work, we will be happy to include the comparison with the FedGDA-GT algorithm from Sun et al. (2022) in our experiments.
>
> ---
> **Final Comments**:
> Thank you again for the review and efforts in our paper. Hopefully, we have managed to address all your concerns. **Please consider raising your score if you agree. If you believe this is not the case, please let us know so that we have a chance to respond.** Thank you.

---

> > ### Comment · Reviewer_HQew · 2023-11-23
> >
> > Thanks for your response, it has addressed some of my concern. The authors have added numerical experiments in comparison with FedGDA-GT and highlighted their novelty in the analysis. I will be happy to raise my score to borderline accept.

---

### Official Review · Reviewer_5mwf · 2023-10-30

**Soundness:** 2 fair
**Presentation:** 2 fair
**Contribution:** 2 fair
**Rating:** 5
**Confidence:** 3

**Summary:**

This paper studies minimax optimization and variational inequality problems in a distributed setting. It proposes and analyzes several novel local training algorithms under a single framework for solving a class of structured non-monotone VIPs.

**Strengths:**

1. The minimax problem covers a variety of crucial tasks within the field of machine learning, and it becomes imperative to investigate minimax problems in a distributed context.

2. It proposes a single framework for solving a class of structured non-monotone VIPs

**Weaknesses:**

1. The main concern is the motivation behind this paper is not clear. Why do we need to design communication-efficient federated learning algorithms suitable for multi-player game formulations


2. The experimental analysis is very limited.

3. The main contributions of this paper are theoretical analysis. It would be better if this paper were submitted to other conferences such as COLT, and AISTAT.

**Questions:**

1.  minimax optimization is important. But do GAN, adversarial training, robust optimization and multi-agent reinforcement learning tasks match the study in this paper?

---

> ### Author Response · Authors · 2023-11-22
> **Response to Reviewer 5mwf**
>
> Thank you very much for your time. We appreciate your effort in reviewing.
>
> Thanks also for acknowledging that the minimax problem covers a variety of crucial tasks within the field of machine learning, and it becomes imperative to investigate minimax problems in a distributed context. We also appreciate that you highlight as a strength our proposed single framework for solving a class of structured non-monotone VIPs.
>
> ---
> **Question 1**: The main concern is the motivation behind this paper is not clear. Why do we need to design communication-efficient federated learning algorithms suitable for multi-player game formulations
>
> **Response 1**: Thank you for the question. Let us further explain the motivation of our study.
>
> As we mentioned in the introduction of our manuscript, in the last few years, more and more machine learning models have moved from the classical optimization formulation to a multi-player game perspective where the problems are formulated as the equilibrium of a game. Also, considering communication-efficient federated learning algorithms are natural with the surge of data-intensive applications and factors like privacy. On that end, several applications require solving min-max optimization problems and VIPs using federated learning algorithms. Recent studies have been conducted on the combination of federated learning approaches and multi-player games. For example, paper [1] (see below) focuses on federated generative adversarial networks (GANs), while the work [2] explains how federated learning algorithms can be used in adversarial learning. Both problems are classical applications formulated as min-max problems (a special case of VIPs) that can also be cast as special cases of our general framework. This clearly motivates the reasons behind working in this area. In addition, existing work on the convergence analysis cannot showcase the superiority of Local GDA-type algorithms, or they need some other strong assumptions like bounded heterogeneity. Regarding all the arguments above, we believe our study here is fully motivated, and our result is significant to the community.
>
> [1] Rasouli, Mohammad, Tao Sun, and Ram Rajagopal. "Fedgan: Federated generative adversarial networks for distributed data." arXiv preprint arXiv:2006.07228 (2020).
>
> [2] Zhang, Jie, et al. "Delving into the adversarial robustness of federated learning." arXiv preprint arXiv:2302.09479 (2023).
>
> ---
> **Question 2**: The experimental analysis is very limited.
>
> **Response 2**: Our work primarily focuses on the theoretical analysis of the proposed novel algorithms. As we mentioned in our manuscript with our experiments, our goal was to corroborate our theoretical findings. That is, running experiments on very complicated problems is not the main focus of our work. The selected experiments are well-designed, and through them, we show the performance of our algorithms for solving **robust learning** problems on **real datasets**. This is a practical setting satisfying the assumptions of our setting.
>
> Per the suggestion of Reviewers bdRp and  HQew, we tested the proposed methods in different settings (not captured by our theory) and compared them with other recently proposed methods, and for all settings, our approach has favorable performance.
> For example, in the burglar problem, our proposed method outperforms Local GDA and Local EG. In addition, for all settings tested, our method is faster than the recently proposed FedGDA-GT. Please see the figures attached in our general response at the beginning of our rebuttal and the individual responses to Reviewers bdRp and  HQew. As we show, our algorithm performs better compared to other state-of-the-art algorithms in the problems we considered.
>
> ---
> **Question 3**: The main contributions of this paper are theoretical analysis. It would be better if this paper were submitted to other conferences such as COLT, and AISTAT.
>
> **Response 3**: We politely disagree with this statement. In fact, we believe that ICLR is embracing interdisciplinary works covering both theory and experimental applications, and our paper bridges the gap between theoretical analysis and practical applications. This fusion of theoretical depth and practical relevance aligns closely with the mission of ICLR. Furthermore, our work has significant ties to real-world applications. The contributions presented in our paper are not just novel; they also hold the potential to influence a variety of practical domains. Given this, we consider ICLR a fitting venue for our work as COLT and AISTATS.
>
> ---
> **Final Comment**:
> Thanks again for your review and efforts in our work.
>
> **If you agree that we have addressed all the issues, please consider raising your score. If you believe this is not the case, please let us know so that we have a chance to respond. Thank you.**

---

> > ### Comment · Reviewer_5mwf · 2023-12-02
> >
> > Thanks a lot for the reply.  The experimental analysis in this paper is very limited and we cannot verify the theoretical analysis from experiments results. I encourage authors to add more experiments to match and verify your analysis. I will keep my score.

---

### Official Review · Reviewer_bdRp · 2023-11-02

**Soundness:** 3 good
**Presentation:** 4 excellent
**Contribution:** 3 good
**Rating:** 6
**Confidence:** 4

**Summary:**

This paper suggests a family of algorithms for decentralized and federated learning settings of problems of solving variational inequalities. This algorithms share the same framework which is based on ProxSkip optimisation method. Concise convergence guarantees are obtained for these methods with general assumptions on problem, and obtained convergence rates expectably improve previously existed algorithms, because achieve acceleration and variance reduction.

**Strengths:**

All the proofs are concise and easy to follow, which is a metodical merit of the paper. Though the improvements achieved are not surprising, the way they were achieved is demonstrative enough.

**Weaknesses:**

Experiments seem not comprehensive. It would be interesting for authors to consider more complicated variational inequality problems than random quadratic and least squares, for example, particular test matrix games like policeman and burglar problem https://www2.isye.gatech.edu/~nemirovs/BrazilTransparenciesJuly4.pdf, and advanced practical problems like GAN training. Also, figures with comparison of the methods have onlu convergence curves without shadows showing standard deviation of function values from run to run, which is required due to stochasticity of the algorithms.

**Questions:**

1) Is usage of colors in Appendix C motivated?
2) Only strongly-monotone co-coercitive case is considered in the paper. Can authors report on the convergence guarantees in non-strongly-monotone case?

---

> ### Author Response · Authors · 2023-11-22
> **Response to Reviewer bdRp**
>
> Thank you very much for your time, and we appreciate your effort in reviewing our work.
>
> We are glad that the Reviewer bdRp finds the proofs of our results concise and easy to follow. The main concerns of bdRp are related to our paper's experiments. We followed your suggestions to update the numerical experiments. We have added the details to your concerns below:
>
> ---
> **Question 1**: Experiments seem not comprehensive. It would be interesting for authors to consider more complicated variational inequality problems than random quadratic and least squares, for example, particular test matrix games like policeman and burglar problem, and advanced practical problems like GAN training.
>
> **Response 1**: The main goal of our experimental section was to corroborate our theoretical findings. As such, we selected popular experiments that satisfy all of our assumptions.
> We thank the reviewer for the suggestion. Following Reviewer bdRp's suggestion, we used our method to solve the policemen burglar problem, even though this does not satisfy our settings perfectly (it is a bilinear problem).
>
> [(HERE)](https://ibb.co/DKY0LYw) is the result based on our fine-tuned stepsize in the deterministic setting. In the attached figure, we plot the duality gap (on the y-axis) with respect to the moving average of the iterates, i.e., $\frac{1}{K} \sum_{k = 1}^K x_k$ (here, $x_k$ is the output after $k$ many communications). Note that the duality gap for the problem $\min_{x \in \mathcal{X}} \max_{y \in \mathcal{Y}} f(x, y)$ at $(\hat{x}, \hat{y})$ is defined as $\max_{y \in \mathcal{Y}} f(\hat{x}, y) - \min_{x \in \mathcal{X}} f(x, \hat{y})$. You can see that our proposed algorithm clearly outperforms Local EG and Local GDA. We will happily include the extra experiment in the camera-ready version of our work.
>
> ---
> **Question 2**: Also, figures with comparison of the methods have only convergence curves without shadows showing standard deviation of function values from run to run, which is required due to stochasticity of the algorithms.
>
> **Response 2**: Thank you very much for the suggestion. In our experiments, each stochastic algorithm in the stochastic setting was run for 10 trials, and in the final figures we plotted the mean of the trajectories (still fair and standard comparison in the stochastic regime). We can easily include the suggestion of the reviewer in the camera-ready version of our work and revise the figures to include shadows showing the standard deviation of the stochastic algorithms. Example for the updated plots for  Figure 1(b) and 1(d) is inlcuded [(HERE)](https://ibb.co/Rcr8S3K) and [(HERE)](https://ibb.co/RbNqLTk), respectively. We plan to include such plots in the camera-ready version of the paper.
>
> ---
> **Question 3**: Is usage of colors in Appendix C motivated?
>
> **Response 3**: We simply mark the key variables in the algorithm using colors aiming to facilitate the readability of the paper.
>
> ---
> **Question 4**: Only strongly-monotone co-coercitive case is considered in the paper. Can authors report on the convergence guarantees in non-strongly-monotone case?
>
> **Response 4**: Thank you for the question. Let us highlight that our theorems hold quasi-strongly monotone and star-cocoercive problems. This is a structured, non-monotone class of problems, which, as we explained in the paper, includes several non-monotone problems (non-convex, non-concave min-max problems) as special cases.
>
> Extending the convergence guarantees of our proposed methods to different settings is an interesting and challenging future research direction. Let us also note that even in the much simpler minimization problems (single-player), no convergence guarantees with an accelerated communication complexity have been provided beyond the strong convex case.
>
> Following the reviewer's suggestion in the first question, we have applied our algorithm in the Policemen Burglar Game, and as shown in the attached figure above, our algorithm still outperforms the popular methods of Local GDA and Local EG. This gives the motivation to dive into the theoretical understanding of the methods in different settings beyond the (quasi-)strongly-monotone case.
>
> ---
> **Final Comment**: In our opinion, the issues/weaknesses raised are minor and can be easily handled in the camera-ready version of our work, as explained above.
> **If you agree that we managed to address all issues, please consider raising your score. If you believe this is not the case, please let us know so that we have a chance to respond.**

---

### Official Review · Reviewer_5WaQ · 2023-11-09

**Soundness:** 3 good
**Presentation:** 3 good
**Contribution:** 3 good
**Rating:** 8
**Confidence:** 4

**Summary:**

This paper provides a unified analysis of algorithms for general regularized VIPs and distributed VIPs. The proposed algorithms improve communication complexity and have strong performance compared to state-of-the-art algorithms. The paper's main contributions include the development of a new communication-efficient algorithm for solving VIPs, theoretical analysis of the algorithm's convergence properties, and experimental results demonstrating the algorithm's effectiveness on a range of problems.

**Strengths:**

Originality: The proposed algorithms are inspired by ProxSkip algorithm with a probability and a control variate. As far as I'm concerned, the originality of this paper is not strong, but the strength lies in the framework that recovers state-of-the-art algorithms and their sharp convergence guarantees.

Quality: The paper's theoretical analysis of the proposed algorithm's convergence properties is rigorous and well-supported. The paper's experimental results demonstrate the algorithm's effectiveness on a range of problems.

Clarity: The paper is well-written and organized, with clear explanations of the key concepts and results.

Significance: This paper has the potential to advance the field of distributed/federated learning and have practical implications for solving variational inequality problems in machine learning.

**Weaknesses:**

1. $prox_{\gamma R}(v)$ in Equation (4) should be $prox_{\gamma R}(x)$.

2. This paper does not mention the challenges or difficulties in algorithm design or theorem proofs. More explanations may help to clarify this paper’s originality.

**Questions:**

1. Theorem 3.1 shows that ProxSkip-VIP converges to the neighborhood of the solution, while the first experiment shows that the proposed variance-reduced method converges to the exact solution. Please add some words to explain it.

2. Is the choice of the probability ($p=\sqrt{\gamma\mu}$) because of the purpose of analysis? What impact will the change of $p$ have on the performance of the proposed algorithms.

---

> ### Author Response · Authors · 2023-11-22
> **Response to Reviewer 5WaQ**
>
> Thank you for investing time in the review process and for the feedback on our work.
>
> We are glad that Reviewer 5WaQ finds our paper well-organized with a clear explanation of concepts and rigorous proofs for convergence guarantees. We also thank the reviewer for acknowledging the potential of our work to advance the field of distributed/ federated learning. Moreover, Reviewer 5WaQ agrees that our experiments explain the effectiveness of the proposed algorithms on a range of problems. We address your concerns as follows:
>
> ---
> **Question 1**: Typo in Equation (4)
>
> **Response 1**: Thank you for the pointer. We have fixed it in the updated version.
>
> ---
> **Question 2**: This paper does not mention the challenges or difficulties in algorithm design or theorem proofs. More explanations may help to clarify this paper’s originality.
>
> **Response 2**: Thank you for the suggestion. We should certainly add the following paragraph to the camera-ready version of our work (where we will have an extra page for the main paper) highlighting the challenges/difficulties in algorithm design and theorem proofs:
>
> **Challenges or difficulties in algorithm design or theorem proofs**:
> The theoretical results of our work and the proposed algorithms are inspired by Mishchenko et al. (2022), who first proposed a ProxSkip algorithm for solving the minimization problem.  However, our work extends these ideas to the more challenging min-max and VIP settings. The detailed proof procedures of our work differ substantially compared to Mishchenko et al. (2022) as we allow the use of a more general unbiased operator via the Key Assumption 2.2. This key Assumption 2.2 on stochastic estimates allows us to study several variants of the ProxSkip update rule under a single framework, including the stochastic (Corollary 3.5), the deterministic (Corollary 3.6), and a variance-reduced method (Corollary 3.9).
>
> In addition, our convergence analysis requires using the quantity $M\gamma^2 \sigma_t^2$ in the Lyapunov function appearing in our main theorems, which also made the convergence guarantees of our work much different. Finally, for our convergence results, we needed to use recent convergence tricks from the analysis of gradient descent-ascent (GDA) from Beznosikov et al. (2022b) in combination with the ProxSkip ideas, which was also more challenging.
>
> An important point that is also good to have in mind is that the previous works in simple optimization problems (Mishchenko et al. (2022) and Malinovsky et al. (2022)) involve function suboptimality (a concept that cannot be useful in VI problems) in the assumptions and their proof techniques. In our setting and proofs, there are no function values. Thus, the difference between the two pieces of literature does not solely lie in different settings (minimization vs VIPs) but begins at a deeper conceptual level.
>
> ---
> **Question 3**: Theorem 3.1 shows that ProxSkip-VIP converges to the neighborhood of the solution, while the first experiment shows that the proposed variance-reduced method converges to the exact solution. Please add some words to explain it.
>
> **Response 3**: We agree with the reviewer that the general Theorem 3.1 shows that ProxSkip-VIP (with any unbiased estimator) converges to the neighborhood of the solution.  However, we highlight in Lemma 3.8 that for the special case of a variance-reduced variant, the corresponding parameters $D_1=D_2=0$, which in turn drives the result in Theorem 3.1 to converge exactly to the solution. This is why the variance-reduced method converges to the exact solution in the first experiment. The experiment verifies precisely our theoretical findings.
>
> ---
> **Question 4**: Is the choice of the probability ($p=\sqrt{\gamma\mu}$) because of the purpose of analysis? What impact will the change of $p$ have on the performance of the proposed algorithms.
>
> **Response 4**: Thank you for the interesting question. The general result of Theorem 3.1 holds under any choice of probability $p$. However, to obtain the accelerated communication complexity in Corollary 3.2, the choice $p=\sqrt{\gamma\mu}$ is needed. As the reviewer pointed out, this is an artifact of the proof techniques related to our theoretical analysis.
> We also agree that different choices of $p$ will lead to different convergence guarantees for the proposed algorithms.
> This requires a detailed check on each different choice of $p$. For example, if $p$ is set to be a small enough choice satisfying $\tau=p^2$, via our theory, we can show that the stepsize $\gamma$ is required to satisfy
> $
> \frac{p^2}{\mu}\leq \gamma\leq \sqrt{\frac{p^2\epsilon}{2(D_1+MD_2)}}.
> $
> Thus, in some particular cases of parameter settings, there will be no feasible $\gamma$ that leads to convergence.
> We highlight that our proposed parameter choice is universally applicable (there are always choices of all parameters that guarantee convergence).

---

> > ### Author Response · Authors · 2023-11-22
> > **Response to Reviewer 5WaQ (cont.)**
> >
> > ---
> >
> > **Final Comments**:
> >
> > Thanks again for the positive evaluation and recognition of our work.
> > In our opinion, all issues raised are minor, not flaws, of our work. That is, they can be easily handled in the camera-ready version of our work.
> >
> > **If Reviewer 5WaQ agrees with us that these are minor corrections, we hope they can consider updating their assessment to support our work.**

---

> > ### Comment · Reviewer_5WaQ · 2023-11-22
> > **Response to rebuttal**
> >
> > Thanks for addressing my concerns. The explanations about the challenges in algorithm design and theorem proofs help a lot to clarify this paper’s novelty. I will be raising my score.

---

### Author Response · Authors · 2023-11-22
**Thanks to all Reviewers**

We thank the reviewers for their valuable feedback and time.

In particular, we appreciate that the reviewers acknowledged the following strengths of our work:
- Reviewer **5WaQ, bdRp** finds the proofs concise and rigorous, the presentation of the paper is well-written, also Reviewer **HQew** appreciates the convergence rates provided in our work.
- Reviewer **5WaQ** further acknowledges the significance of our work in terms of the potential to advance the field of distributed/federated learning.
- Reviewer **5mwf** finds the unified framework for solving a class of structured non-monotone VIPs interesting and recognizes our work's importance for studying min-max problems in the distributed setting.
- Reviewer **HQew** also appreciates the variance reduction variant of our algorithm.

With our rebuttal, we address all raised issues. **Here we highlight again that with our work:**

- First by providing a connection between regularized variational inequality problems (VIPs) and federated learning setting, we provide **the first algorithms** (ProxSkip-VIP-FL and ProxSkip-L-SVRGDA-FL) for solving **federated minimax optimization problems on heterogeneous data** with an improved communication complexity. Our theoretical results hold for the more general VIP setting.
- Second, we provide a **unified theoretical framework for designing efficient local training methods for federated VIP problems** by using more relaxed assumptions capturing several stochastic variants of the gradient descent-ascent method.
- Via our proposed **unified framework, we can recover the best-known rates for well-known methods** like Proximal SGDA for minimax problems and ProxSkip for composite minimization problems. This highlights the tightness of our analysis. Also we avoid the common **bounded heterogeneity assumption**.
- We corroborate our theoretical results with **extensive experimental testing** on problems satisfying our main assumptions. We show that our proposed algorithms **outperform the existing state-of-the-art distributed algorithms** used to solve the same class of problems.

In our rebuttal, we provide further experiments to demonstrate our approach's benefit, as the reviewers suggested. Let us provide here a summary of the additional experiments. We provide more details later in the rebuttal (response to individual reviewers).

Following the suggestions from Reviewers **bdRP** and **HQew**, we extended our experimental evaluation beyond our previous setting and attained new results:

- First, we focus on the policemen burglar problem suggested by **bdRP** in the deterministic setting, and we include the results [(HERE)](https://ibb.co/DKY0LYw). In our experiment, we fine-tuned the stepsizes for all methods. In the attached figure, we plot the duality gap (on the y-axis) of the moving average of the iterates. Our algorithm clearly outperforms Local EG and Local GDA in this setting. This result showcased the benefits of the proposed algorithms in the non-(quasi)-strongly-monotone cases, we leave the detailed theoretical analysis of this setting as a future direction of our work.
- Second, we have incorporated the FedGDA-GT algorithm from Sun et al. (2022) as suggested by **HQew** into the comparison of our method with prior work. We attach the new result [(HERE)](https://ibb.co/PMwcfwT), for the deterministic strongly-monotone quadratic game (Eq.6). In comparison to the algorithm from Sun et al. (2022), our method avoids the compact constraints in the function. In addition, our algorithms and convergence guarantees hold also in the stochastic case. Finally, we highlight that the new experiment shows that our proposed algorithm (ProxSkip-GDA-FL)  outperforms all other candidates, including FedGDA-GT, which further rationalizes the empirical privilege of our approach.

**We hope that you will engage with us in a back-and-forth discussion and we will be most happy to answer any remaining questions.** Thank you very much.

---

### Meta-Review · Area_Chair_2db8 · 2023-12-07

**Metareview:**

This paper studies GDA for distributed VIP. This setting is motivated by applications of multi-player games on distributed networks, where the problem cannot be modeled as a classical optimization problem and VIP is needed. Inspired by earlier works on federated learning, the authors proposed a local GDA for solving distributed VIP with improved communication complexity. Both the problem and the results are important.

**Justification For Why Not Higher Score:**

Results not significant enough.

**Justification For Why Not Lower Score:**

It has interesting results and should be accepted.

---

### Decision · Program_Chairs · 2024-01-16

Accept (poster)